# Endonucleosis mediates internalization of cytoplasm into the nucleus

Ourania Galanopoulou[1,2], Evangelia C. Tachmatzidi[1,2], Elena Deligianni[1], Dimitris Botskaris[1,2], Kostas C. Nikolaou[3], Sofia Gargani[3], Yannis Dalezios [4], Georges Chalepakis[2] & Iannis Talianidis [1] ✉

Setd8 regulates transcription elongation, mitotic DNA condensation, DNA damage response and replication licensing. Here we show that, in mitogen-stimulated liver-specific Setd8-KO mice, most of the hepatocytes are eliminated by necrosis but a significant number of them survive via entering a stage exhibiting several senescence-related features. Setd8-deficient hepatocytes had enlarged nuclei, chromosomal hyperploidy and nuclear engulfments progressing to the formation of intranuclear vesicles surrounded by nuclear lamina. These vesicles contain glycogen, cytoplasmic proteins and even entire organelles. We term this process "endonucleosis". Intranuclear vesicles are absent in hepatocytes of Setd8/Atg5 knockout mice, suggesting that the process requires the function of the canonical autophagy machinery. Endo-nucleosis and hyperploidization are temporary, early events in the surviving Setd8-deficient cells. Larger vesicles break down into microvesicles over time and are eventually eliminated. The results reveal sequential events in cells with extensive DNA damage, which function as part of survival mechanisms to prevent necrotic death.

Genome integrity is fundamental for life. Numerous environmental and intrinsic stimuli can cause DNA damage, which result in mutations, genome instability, premature aging or cancer. Eukaryotes have evolved a complex set of mechanism that sense and repair DNA damage to safeguard the genome. Mild DNA damage normally leads to cell cycle arrest and repair of the lesions, while severe and irreparable injury results in the activation of cell death programs, such as apoptosis, mitotic catastrophes or necrosis[1,2]. In some cases, cells accumulating massive DNA damage can escape death by entering a senescence stage, characterized by cell cycle arrest, DNA damage, senescence-associated secretory phenotype (SASP) and deregulated metabolism[3,4]. The molecular basis of the decision-making processes that lead to cell death or senescence is not well understood.

Setd8 is one of the most important upstream regulators of genome stability, with pivotal functions in multiple chromatin-templated processes. Setd8 is the sole enzyme catalyzing H4K20 monomethylation[5,6], which is used as substrate by Suv4-20h1/2 to generate H4K20Me$_2$ and H4K20Me$_3$. H4K20 methylation facilitates nucleosomal folding and heterochromatin formation[6]. Previous studies have established the essential role of Setd8 and H4K20 methylation in the regulation of genome integrity[7–10], mitotic chromatin condensation[11,12], replication licensing[13,14] and DNA damage response[6–10]. Exogenous DNA damage initially leads to rapid degradation of Setd8, to allow decompaction and repair of the chromatin in the damaged areas[7,8,11]. However, due to the crucial role of H4K20Me$_1$ in 53BP1 recruitment to DNA damage sites and in chromatin condensation during mitosis, loss of function mutations or sustained depletion of Setd8 leads to extensive accumulation of unrepaired DNA lesions, which leads to cell death in many different cell types[10].

We have previously shown that hepatocyte-specific inactivation of *Setd8*, leads to cell division-dependent extensive DNA damage and

[1]Institute of Molecular Biology & Biotechnology, Foundation for Research and Technology Hellas, Heraklion, Crete, Greece. [2]Dept. of Biology University of Crete, Heraklion, Crete, Greece. [3]Biomedical Sciences Research Center Alexander Fleming, Vari, Greece. [4]School of Medicine University of Crete, Heraklion, Crete, Greece. ✉e-mail: talianid@imbb.forth.gr

massive necrotic cell death[15]. As the majority of hepatocytes in the adult liver are in the G0 phase, necrotic cell death in 2–4 months old mice is initially observed in microscopic focal areas of the liver. The initial cell death triggers a local regenerative process, whereby neighboring cells enter the cell cycle and accelerate the death of the existing Setd8-deficient hepatocytes, resulting in the expansion of the necrotic zone. Within the necrotic areas of the liver, we observed enlarged hepatocytes and accumulation of inflammatory cells[15].

Large hepatocytes appeared homogenously in all areas of the liver after either partial hepatectomy, which induces cell cycle entry[15], or under metabolic stress conditions, such as fasting, high fat diet or pyruvate challenge[12]. At 8–12 months of age, Setd8-deficient mice develop full-blown spontaneous liver tumors comprising transformed hepatocytes with cancer stem cell properties[15–17].

The apparent discrepancy between Setd8 deficiency-induced necrosis and the late-onset hepatocellular carcinoma prompted us to further investigate and characterize the process of hepatocyte enlargement triggered by spontaneous or induced cell proliferation.

## Results

### Internalization of cytoplasm into the nuclei by endonucleosis

Treatment of liver-specific *Setd8* knockout mice (Setd8-LKO) with TCPOBOP, a widely used CAR agonist that induces robust hepatocyte proliferation[18–20], resulted in synchronous and homogeneous enlargement of hepatocytes within 24 h and an ~8-fold increase in the nuclear size (Fig. 1a and Supplementary Fig. 1a). Enlarged hepatocytes occupied almost the entire liver space along with a significant enrichment of small mononuclear cells (Fig. 1a and Supplementary Fig. 1c–e). Indicative of the massive cell death in Setd8-LKO livers, only about 10% of the hepatocyte population remained in Setd8-LKO livers compared to wild type mice (Supplementary Fig. 1b).

The vast majority, essentially all (>99%) of the enlarged nuclei, contained heterogeneously-sized vesicles, externally coated by Lamin A/C and Lamin B1 (Fig. 1a and Supplementary Fig. 2a). The presence of Sun2, a core component of the inner nuclear membrane-associated LINC complex, that connects the nuclear lamina to actin cytoskeleton[21] and the nuclear pore complex component Nup98[22] in the borders of the vesicles, further confirmed that the membranes coating the vesicles originated from the nuclear envelope (Supplementary Fig. 2c, d). Immunostaining for VAP-A, a marker of the outer nuclear membrane, revealed signals coating the inner side of the vesicles, suggesting that the process generating the vesicles involves the outer nuclear membrane as well (Supplementary Fig. 2e). The latter notion is further supported by the detection of 3 different ER marker proteins, Calnexin, PD1 and ERp72, at the interior of the vesicles (Supplementary Fig. 2f–h). Lamin A/C polymers were thinner but had proportionally higher staining intensity in Setd8-LKO cells than in wild type cells, as revealed by measuring the thickness and the staining intensity of the Lamin A/C signal in a multitude of regions (Supplementary Fig. 3a, b). These data point to a high level of elasticity of the nuclear lamina with stretched and denser lamin polymers in Setd8-LKO hepatocytes. Although the Lamin A/C protein content may not differ locally, its overall levels per individual cell were greatly increased in Setd8-LKO hepatocytes, considering the greatly increased nuclear size and the presence of the numerous intranuclear lamin-coated vesicles. 3D reconstruction of confocal sections demonstrated that the vesicles were positioned entirely within the nuclear space (Fig.1b and Supplementary Fig. 3c). The intranuclear vesicles contained cytoplasmic proteins, such as albumin, calnexin, α-tubulin, F-actin filaments and glycogen (Fig. 1c–g), but excluded nuclear proteins, such as HNF4α (Supplementary Fig. 2b) and DNA (Fig. 1a, b).

Transmission electron microscopy analyses captured several steps of the cytoplasm internalization process. We observed micro-extensions of the nuclear membrane, membrane invaginations towards the nuclear interior and cytoplasm internalization via fusion of the inwardly-folded nuclear membrane. Fusion occurred at the tip of the engulfed areas, thus sequestering cytoplasmic materials into intranuclear vesicles (Fig. 2a, b). As a result of the membrane folding towards the nuclear interior, lamina-associated chromatin is positioned at the external side of the resulting vesicles (Fig. 2a–c). Moreover, in larger vesicles, we could detect glycogen or even whole cytoplasmic organelles, such as endoplasmic reticulum, mitochondria, peroxisomes and autophagosome structures (Fig. 2c). We propose to call this process "endonucleosis", based on its similarity to endocytosis, the process by which external substances are internalized into the cells[23].

The endonucleosis phenomenon is not limited to Setd8-LKO hepatocytes under replication stress. Our previous work examining Setd8-LKO mice at postnatal day 45 demonstrated that, metabolic reprogramming in Setd8-deficient mice generates a highly sensitized state, that upon different metabolic stress conditions, results in cells with highly enlarged nuclei[16]. Here, we tested starvation and Na-pyruvate challenge conditions. The majority of the hepatocytes in the livers of mice fasted for 48 h or those treated with Na-pyruvate for 2 h, contained enlarged hepatocytes with endonucleotic vesicles (Fig. 3a, b).

Endonucleosis was also detected in obese mice in a condition that is independent of Setd8 inactivation. Obesity in the leptin-deficient *ob/ob* mice is associated with fatty livers due to the accumulation of triglycerides in the hepatocytes, which imposes considerable metabolic stress to the cells[24]. In a significant portion (10 to 15%) of the cells we could detect glycogen-containing endonucleotic vesicles using PAS staining and Lamin A/C immunostaining (Fig. 3c, d). Importantly, intranuclear vesicle-containing cells were readily stained for H4K20Me$_1$, demonstrating that these cells expressed functional Setd8 (Fig. 3e). The vesicle-containing cells were also stained positively for γH2A.X, pointing to an extensive accumulation of DNA lesions as a result of lipotoxic stress (Fig. 3e).

In addition, the occurrence of endonucleosis in a wider-range of conditions is further corroborated by the findings of a recent study on *ARID1A* knockout mice, which phenocopies several aspects of Setd8-LKO mice. ARID1A-KO mice develop liver tumors following CAR agonist injection, exhibit excessive accumulation of DNA damage and nuclear abnormalities in some hepatocytes that highly resemble to the nuclear membrane invaginations and nuclear inclusions presented here[25].

### Endonucleosis requires Atg5 function

We have previously reported that constitutive and persistent activation of AMPK in Setd8-LKO mice leads to the appearance of cytoplasmic autophagic structures in parallel to the rise of the lipidated form of Map1lc3b (LC3)[16]. Using GFP-LC3 transgenic mice[26] to monitor LC3 distribution in Setd8-LKO mice, we observed high levels of punctate fluorescence signals within the vesicles, which is characteristic of the activated form of the protein[27], as opposed to the homogenously distributed low levels of nuclear LC3 (Fig. 4a). We also observed a parallel decrease in p62 staining signal, which provides additional evidence for elevated autophagy activity in Setd8-LKO mice (Fig. 4b). The punctate form of LC3 was located inside the vesicles. This form of LC3 is involved in autophagic membrane trafficking and the engulfment of cytoplasmic components into autophagosomes. This strongly suggests that endonucleosis may use the activities of the classical autophagy machinery to internalize cytoplasm into the nucleus. This possibility is supported by our findings in TCPOBOP-treated Setd8/Atg5 double knockout mice, where we observed intermediate-sized nuclei, which lacked endonucleotic vesicles (Fig. 4c–e). Atg5 is an indispensable component of both, the canonical and the non-canonical autophagy process[28]. Thus, we conclude that the function of the cytoplasmic autophagy machinery is required for nuclear membrane folding-mediated endonucleosis.

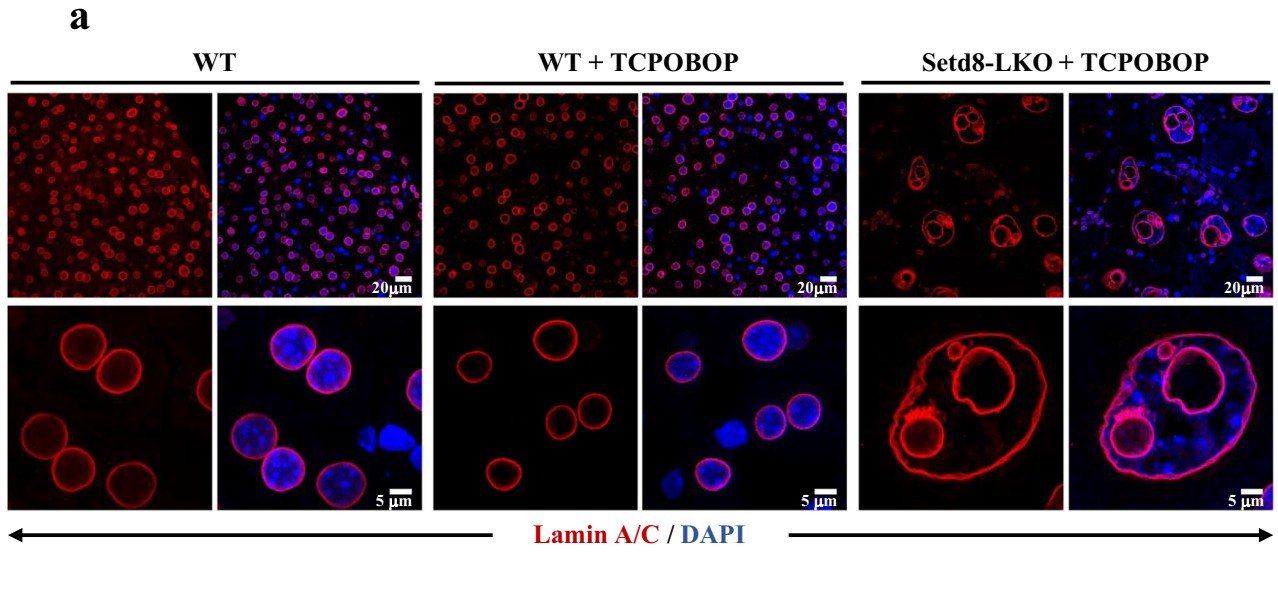

**Fig. 1 | Intranuclear lamin-coated vesicles containing cytoplasmic proteins in Setd8-LKO hepatocytes. a** LaminA/C antibody immunostaining of liver sections from wild type (WT) mice and from wild type or Setd8^lox/lox/AlbCre (Setd8-LKO) mice 24 h after treatment with TCPOBOP (WT + TCPOBOP and Setd8-LKO + TCPOBOP). Fluorescent images of the cells are shown at different magnifications. **b** Orthogonal views of confocal z-stacks showing 3D morphology of internal vesicles in 3 typical Lamin A/C-stained nuclei of TCPOBOP-treated Setd8-LKO hepatocytes. **c**–**f** Representative immunofluorescence images with antibodies against albumin (**c**), calnexin (**d**), α-tubulin (**e**) and F-actin (**f**). **g** Representative image of periodic acid-Schiff (PAS) staining of glycogen.

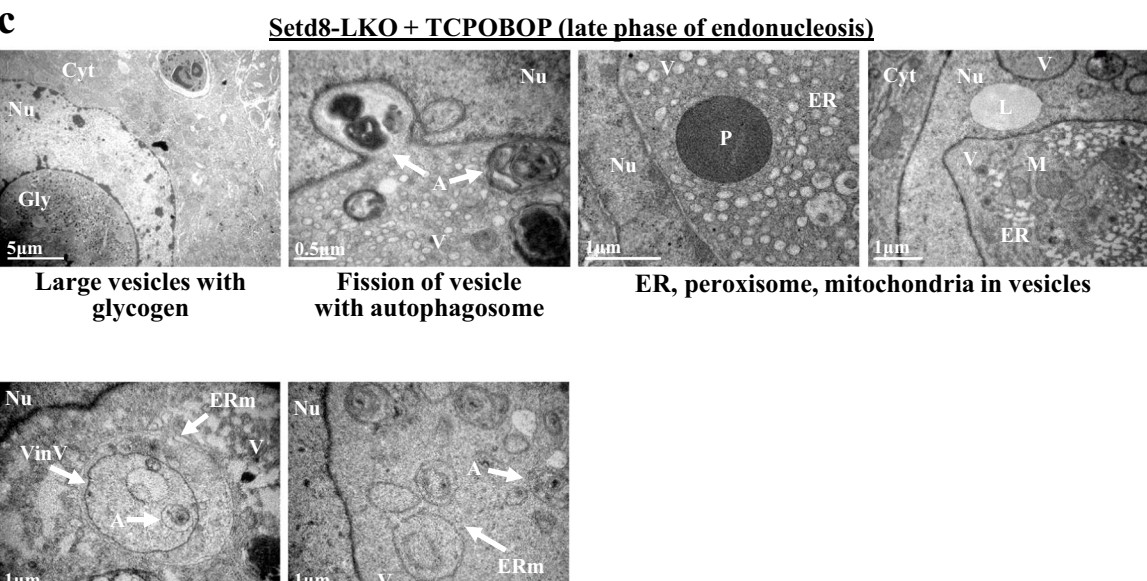

**Fig. 2 | Nuclear membrane folding-mediated internalization of cytoplasmic components. a–c** Transmission electron microscopic images of liver sections from wild type (WT) and Setd8-LKO mice after 24 h of TCPOBOP treatment. Nu nucleus, Cyt cytoplasm, ER endoplasmic reticulum, M mitochondria, V vesicles, Chr chromatin, Gly glycogen, A autophagosome structures, L lipid droplet, VinV vesicles within vesicle, ERm ER membrane.

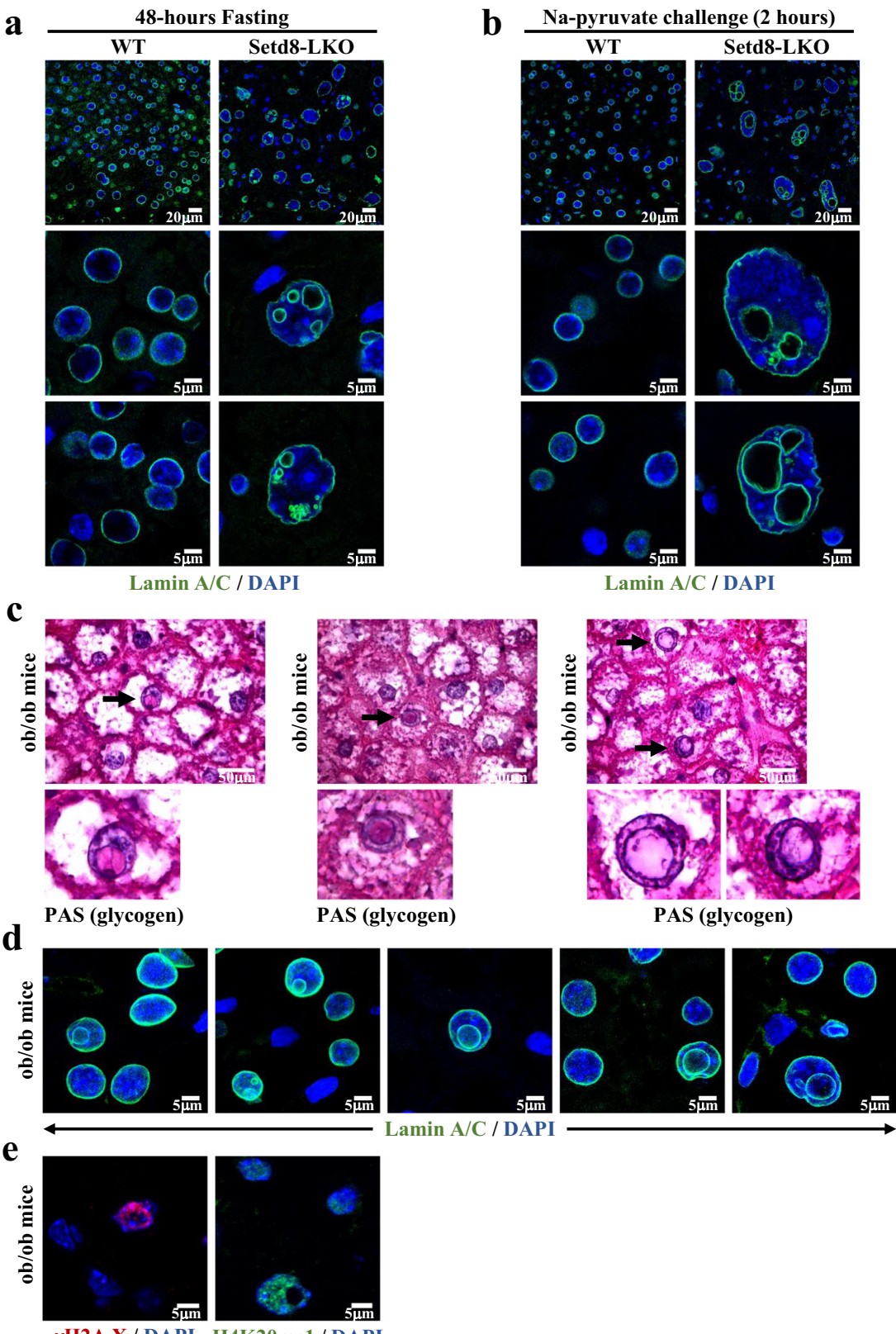

**Fig. 3 | Endonucleosis in different metabolic stress conditions in Setd8-LKO and in ob/ob mice. a, b** Setd8-LKO mice were either fasted for 48 h (**a**) or treated with 2 g/kg Na-pyruvate for 2 h (**b**). Representative images of immunostainings with Lamin A/C antibody are shown in different magnifications. **c** Representative images of Periodic-acid Schiff (PAS) staining of liver sections of *ob/ob* mice. Arrows indicate the presence of glycogen-containing intranuclear vesicles in several hepatocytes. **d** Representative images of liver sections from *ob/ob* mice stained with Lamin A/C antibody. Note the presence of endonucleotic vesicles in several nuclei. **e** Representative images of endonucleotic vesicle-containing hepatocytes in liver sections of *ob/ob* mice stained with γH2A.X (left panel) or H4K20Me₁ (panel at right) antibodies. Note the presence of γH2A.X signal only in nuclei containing endonucleotic vesicles and the presence of H4K20Me₁ signal in all of the hepatocytes of *ob/ob* mice.

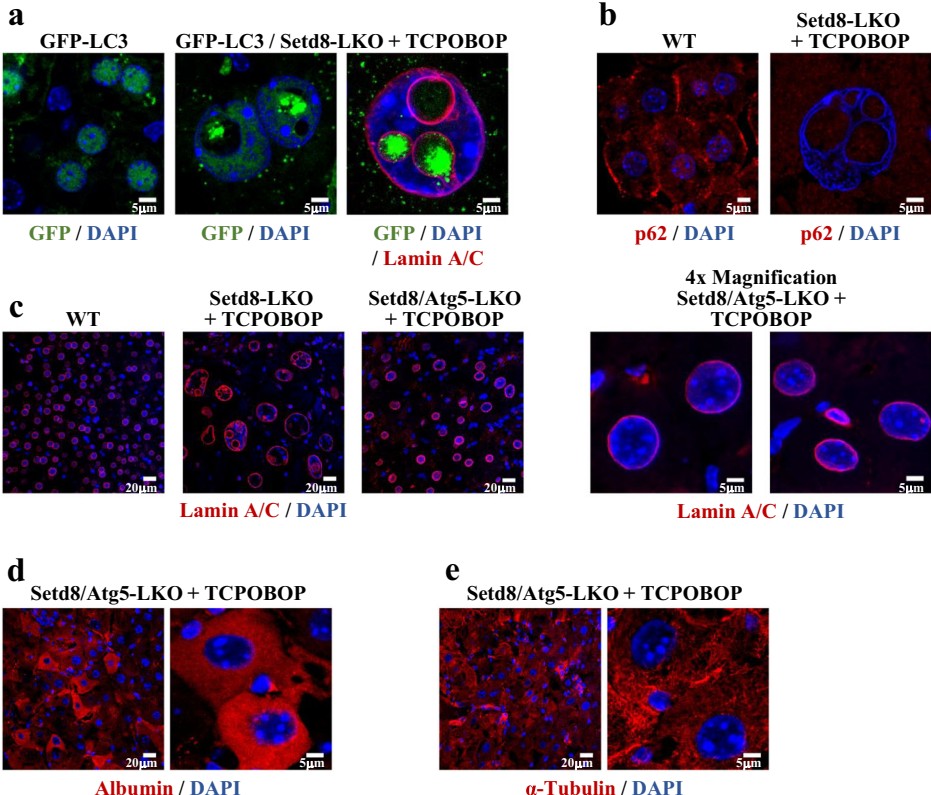

**Fig. 4 | Endonucleosis requires the autophagy regulator Atg5. a** Detection of the autophagy marker protein LC3 by GFP fluorescence in GFP-LC3 transgenic mice and GFP-LC3/Setd8-LKO mice 24 h following TCPOBOP treatment. **b** Immunofluorescence staining for the autophagy receptor protein p62. **c** Lamin A/C staining in wild type, Setd8-LKO and Setd8-LKO/Atg5-LKO double knockout mice after 24 h of TCPOBOP treatment. Note the lack of intranuclear vesicles and the lesser enlargement of the nuclei in the double KO mice. **d**, **e** Immunostaining of TCPOBOP-treated Setd8/Atg5-LKO double knockout hepatocytes for albumin (**d**) and α-tubulin (**e**) demonstrating the lack of endonucleosis-mediated internalization of cytoplasmic proteins in Atg5-deficient cells.

## Hyperploidy, DNA damage and senescence-related features in Setd8-LKO hepatocytes

TCPOBOP induces hepatic cell proliferation by promoting the entry into S phase of hepatocytes, which normally reside at G0 phase. This results in a transient increase of hepatocyte number and liver size, peaking at 36–48 h after treatment[14]. Consistent with previous observations[14,15] in wild type mice, at the early time point of 24 h, we observed a slight increase in the number of cells staining positively for the proliferation marker Ki67, without any changes to the average nuclear size and ploidy of the cells (Fig. 5a, c and Supplementary Fig. 1a). Interestingly, the normally diploid or tetraploid hepatocytes became hyperploid in TCPOBOP-treated Setd8-LKO mice and a significant proportion expressed Ki67 (Fig. 5a, c). Moreover, EdU pulse labeling assays detected a number of cells actively replicating DNA (Supplementary Fig. 4a). Propidium iodide (PI) staining assays showed the loss of diploid cells (with 2 N DNA content), a reduction in the proportion of tetraploid (4 N) cells to 4.5% of the cells and the concomitant appearance of cells with 16 N, 32 N, 64 N, and 128 N DNA content, representing ~95% of the total hepatocyte population (Fig. 5a). Hyperploidy was also confirmed by in situ DNA FISH assays using probes from three different chromosomes (Fig. 5b). The periodicity of the PI staining profile indicates that hyperploidy was generated by consecutive duplication events (endoreduplication) without cell division. This phenomenon could be explained by the role of Setd8 in replication licensing. Previous studies have shown that, in cycling cells, proper recruitment of the origin recognition complex (ORC) to chromatin is mediated by Suv4-2h1/2-catalyzed conversion of H4K20Me1 to H4K20Me2/3-modified nucleosomes[9]. It has been postulated that the regulation and timing of H4K20 mono-, di and tri-methylation is critical for proper replication firing, as decreased H4K20Me1 and increased H4K20Me2/3 levels correlated with aberrant re-replication in Setd8-depleted cells[10]. Consistent with this possibility, we detected high levels of trimethylated and loss of monomethylated H4K20 by immunostaining in TCPOBOP-treated Setd8-LKO hepatocytes (Supplementary Fig. 4b, c).

To examine whether the elevated autophagy activity in Setd8-LKO hepatocytes contributes to hyperploidy, we performed propidium iodide (PI) staining and in situ DNA FISH assays in the livers of Setd8/Atg5 double knockout mice. Consistent with the intermediate-sized nuclei in these mice (Fig. 4c and Supplementary Fig. 1a), we observed loss of cells with 2 N DNA content and polyploidization up to 16 N, with the majority of the cells containing 4 N and 8 N DNA (Supplementary Fig. 5a, b). These results suggest that endoreduplication in Setd8-LKO cells and autophagy-dependent nuclear membrane expansion are coupled, interdependent processes. Alternatively, we speculate that Atg5 may also influence directly DNA replication activity. Supporting this possibility is the finding of a recent study demonstrating the ability of Atg5 to translocate into the nucleus in cells exposed to DNA damaging agents[29].

High levels of endoreduplication are expected to be accompanied by replication stress and genome instability. We used FANCD2 and RPA32 immunostaining to monitor replication stress-induced lesions and γH2AX staining for the detection of double-stranded DNA breaks. We could readily detect a significant number of cells with punctate RPA32, FANCD2 and γH2AX staining patterns in Setd8-LKO hepatocytes (Fig. 5d, e and Fig. 6a). These results are consistent with the reported role of Setd8 in maintaining genome stability[5–10].

The excessive enlargement of Setd8-LKO hepatocytes, the observed cell cycle arrest, and the extensive DNA damage raised the question of whether these cells escape necrotic death by entering a senescent state[3,4]. We could detect a number of hallmark features of

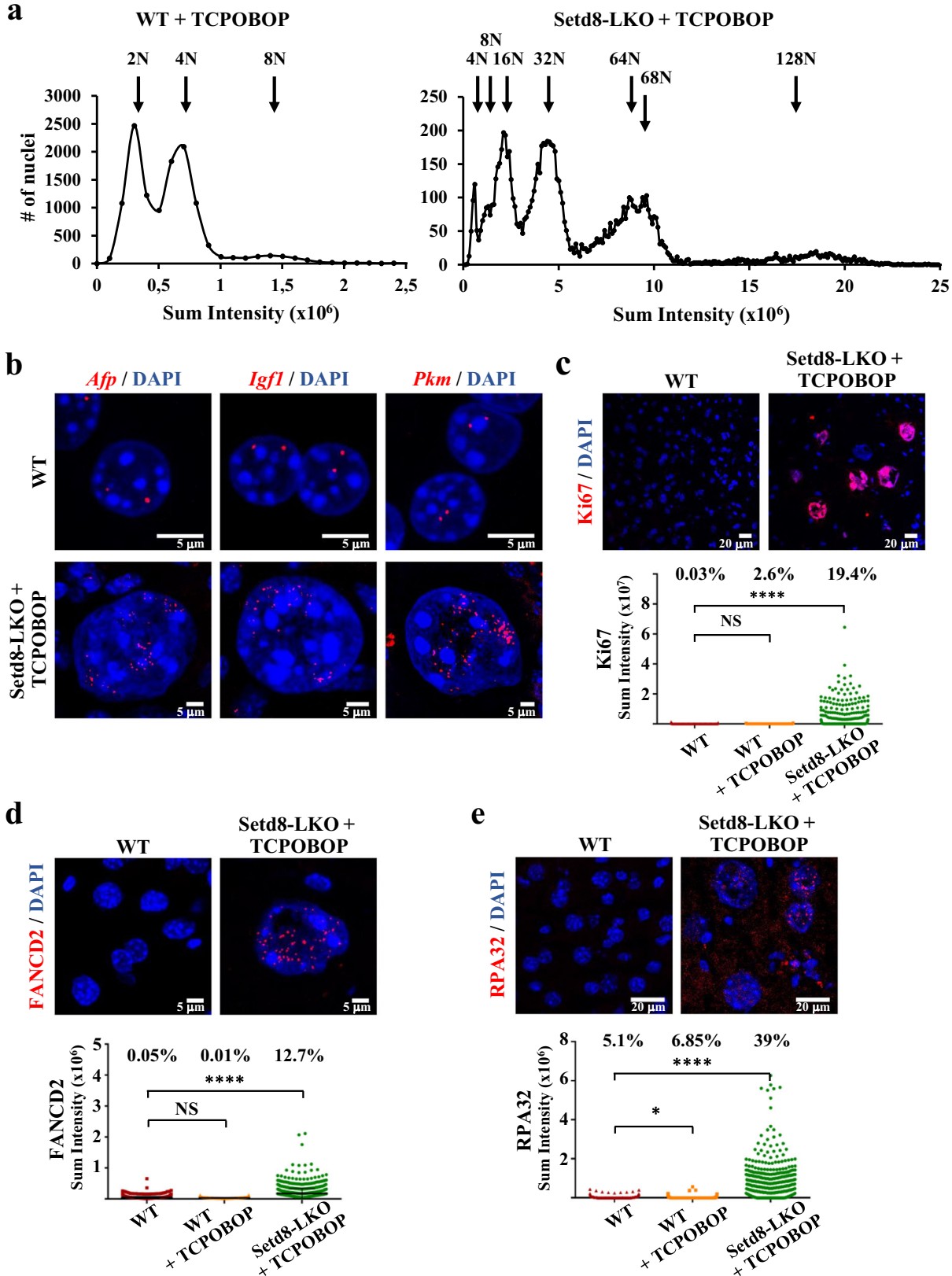

senescence, including high senescence-associated β-galactosidase (SA-β-gal) activity and the exit of HMGB1 from the nucleus in Setd8-LKO hepatocytes (Fig. 6b). Interestingly, SA-β-gal and HMGB1 were also detected in endonucleotic vesicles (Fig. 6b). In addition, H3K9Me₃ and macroH2A staining revealed large focal accumulations, reminiscent of senescence-associated heterochromatin foci (SAHFs) (Fig. 6c).

Importantly, transcriptome analysis of Setd8-LKO cells revealed distinct gene signatures corresponding to hallmark G2/M checkpoint genes, cellular senescence-related and senescence-associated secretory phenotype (SASP) genes (Supplementary Fig. 6a, b, d, e). Significant gene signatures included the group of oncofetal genes (Supplementary Fig. 6c). These genes are highly expressed in

**Fig. 5 | Hyperploidization and replication stress in Setd8-LKO hepatocytes.**
**a** Quantification of nuclear DNA content in TCPOBOP-treated wild type ($n = 12217$) and Setd8-LKO livers ($n = 12163$) by propidium iodide (PI) staining. Graphs show high-content microscopy (HCM) measurements of the PI staining intensity in individual cells after 24 h of treatment. Arrows indicate the chromosomal ploidy of the peaks. The percentage of the cells corresponding to the peak areas were as follows: in WT + TCPOBOP, $2 N = 47.6\%$; $4 N = 44.7\%$, $8 N = 6.7\%$; in Setd8-LKO + TCPOBOP, $4 N = 4.5\%$; $8 N = 5.1\%$, $16 N = 19.5\%$, $32 N = 32.9\%$, $64 N/68 N = 29.5\%$, $128 N = 8.2\%$. **b** Representative DNA FISH images (red) with probes spanning *Afp* (chr5), *Igf1* (chr10) and *Pkm2* (chr9) genes. **c** Immunofluorescence images of liver sections stained with antibody for the proliferation marker Ki67. The graph shows HCM measurements of the Ki67 staining intensities in individual cells ($n = 1161$ for wild type, $n = 1226$ for WT + TCPOBOP and $n = 1228$ for Setd8-LKO + TCPOBOP). Numbers indicate the percentage of cells with staining intensity above threshold. **d-e** Immunofluorescence images of liver sections stained with FANCD2 (**d**) or RPA32 (**e**). The graph shows HCM measurements of the staining intensities in individual cells (FANCD2 staining: $n = 2299$ for wild type, $n = 2334$ for WT + TCPOBOP and $n = 2224$ for Setd8-LKO + TCPOBOP; RPA32 staining: $n = 3298$ for wild type, $n = 3384$ for WT + TCPOBOP and $n = 2458$ for Setd8-LKO + TCPOBOP). Numbers indicate the percentage of cells with staining intensity above threshold. Data in (**d**) are presented as median values (black line) with SD. Data analyses in (**c**, **d** and **e**) were performed using one-way ANOVA test. *$p$-value < 0.01. ****$p$-value < 0.0001. NS, Non-significant. Source data are provided as a Source Data file.

embryonic livers, completely silenced after birth and reactivated in hepatocellular carcinoma, conveying "stemness" properties to cancer cells[30,31]. This latter feature is similar to the recently reported chemotherapy-induced Senescence-Associated-Stemness (SAS) phenotype in lymphomas, which is considered as an early condition triggering escape from cell cycle arrest and driving more aggressive tumor growth[32–34].

Taken together, these results suggest that TCPOBOP-treated Setd8-deficient cells exit the G0 phase and replicate DNA. They become growth-arrested at the next G2 phase of the cell cycle but continue DNA replication, without cell duplication, which results in hyperploidy. These cells enter a specialized state, displaying several hallmark features of senescence, including metabolic reprogramming[16], cell cycle arrest, SASP transcriptome profiles, SAHF and SAS features. Nonetheless, they express LaminA/C and B, and retain replication capacity for some period, which distinguishes them from traditionally defined senescent cells[3,4].

### Partially preserved chromatin domains in hyperploid cells

In mammals, distinct regions of the genome are tightly associated with the nuclear lamina. These large Lamina-Associated Domains (LADs) play important role in heterochromatin-mediated gene silencing and the overall 3D organization of the genome[35,36]. The increased ploidy and the expanded nuclear membrane surface at the nuclear periphery and at the endonucleotic vesicles raise the possibility that LAD distribution and chromatin domains maybe affected in TCPOBOP-treated Setd8-LKO cells. CUT&Tag assays using a Lamin A/C antibody revealed minor changes in LAD distribution compared to wild type cells (Fig. 7a, c), suggesting that the same specific genomic regions interact with nuclear envelope-associated or vesicle-associated lamins even in the hyperploid state.

Interestingly, we found a significant overlap between H4K20Me3-decorated heterochromatin and LADs (Fig. 7a, b). H4K20Me3-decorated heterochromatin in TCPOBOP-treated Setd8-LKO cells remained stable in most regions but a significant number of new sites were also detected (Fig. 7d). The association of H4K20Me3 with nuclear lamins at the inner nuclear envelope and the outer membrane of the internalized vesicles was further validated by proximity ligation assays (PLA) (Supplementary Fig. 7).

Most of the open chromatin areas detected in the wild type genome, including H3K27-acetylated nucleosomes and transposase-accessible regions were unaffected in TCPOBOP-treated Setd8-LKO cells (Supplementary Fig. 8a,b). Despite the number of new locations identified in Setd8-LKO cells, the overall patterns and the periodic clustering of the H3K27ac-decorated, transposase-accessible euchromatin domains and those of H4K20Me3-occupied, lamina-associated heterochromatin domains were found to be largely preserved (Supplementary Fig. 8c).

Examination of the binding locations of the architectural transcription factor CTCF[37,38] and the cohesion ring complex component Smc3[39] revealed a number of new sites in the hyperploid Setd8-LKO hepatocytes compared to wild type cells. The new CTCF and Smc3 locations outnumbered those detected in wild type cells by 3-fold and 1.2-fold, respectively (Supplementary Fig. 9a and 10a, d). However, the majority of the new CTCF locations did not possess a canonical CTCF binding sequence motif and had lower binding strength and a more diffuse distribution (Supplementary Fig. 9b, c). Smc3 binding in the new locations had a lower affinity and a greatly altered distribution relative to CTCF-occupied putative loop-anchor positions (Supplementary Fig. 10b, c). The results suggest that in Setd8-LKO hepatocytes, CTCF and Smc3 associate with genomic sites observed in wild type cells (constant peaks) but also bind non-specifically to a large number of other regions, probably due to the presence of an abnormally high number of chromosomes. Given the low levels of co-occupancy at canonical CTCF locations, we estimate that the new associations are probably unstable and non-functional.

### Reversal of endonucleosis and hyperploidy

Despite the abnormal phenotype of TCPOBOP-treated Setd8-LKO mice, with a 90% loss of the initial hepatocytes, the mice were viable for a long period of time after the single TCPOBOP treatment. In fact, similarly to the untreated Setd8-LKO mice[15], they developed spontaneous liver cancer at 6–8 months of age. We therefore examined the cells beyond the 24 h time point, for a period of 65 days after the single TCPOBOP injection. On days 7 and 10 we observed a significant reduction in the number of cells containing large vesicles and the parallel appearance of smaller vesicles (Fig. 8a). The smaller vesicles became predominant in most cells on 10 or 20 days after treatment, and accumulated along the nuclear envelope. Importantly, the smaller vesicles did not contain albumin or α-tubulin at these time points, suggesting that cytoplasmic proteins were degraded (Fig. 9a). Twenty and 30 days after treatment, numerous cells had nuclear membrane blebbings wrapped internally by lamina and detachments of DNA-containing structures into the cytoplasm (Fig. 8a). Thirty to 65 days after treatment, intranuclear vesicles were essentially absent (i.e. we observed <1 cell with vesicles per 50 fields) and the most common features were segregating nuclei, without mitotic structures, which could be detected by the patterns of DAPI and α-tubulin staining (Figs. 8a, 9a). Close-up images captured fission of large intranuclear vesicles into microvesicles (Fig. 8b), line-up of microvesicles along the nuclear envelope (Fig. 8c, d), fusion of microvesicles with nuclear membrane exposing the vesicle interior towards the cytoplasm (Fig. 8e, f), nuclear membrane blebbing and detachment of small nuclear membrane-coated DNA-containing vesicles (Fig. 8g, h) and various stages of nuclei segregating into smaller-sized ones (Fig. 8i, j). In parallel, we observed a significant reduction in the average nuclear size and an increase in the overall number of hepatocytes per field (Fig. 9c, e). The PI staining patterns of the nuclei demonstrated a significant reduction in ploidy, and indications for the appearance of cells with aneuploidy (Fig. 9b). Ki67-positive cells were greatly reduced and essentially disappeared by day 30 after TCPOBOP treatment (Fig. 9d).

Taken together, the above results suggest that the processes leading to increased nuclear size, hyperploidy, and accumulation of endonucleotic vesicles in the cells are reversible processes.

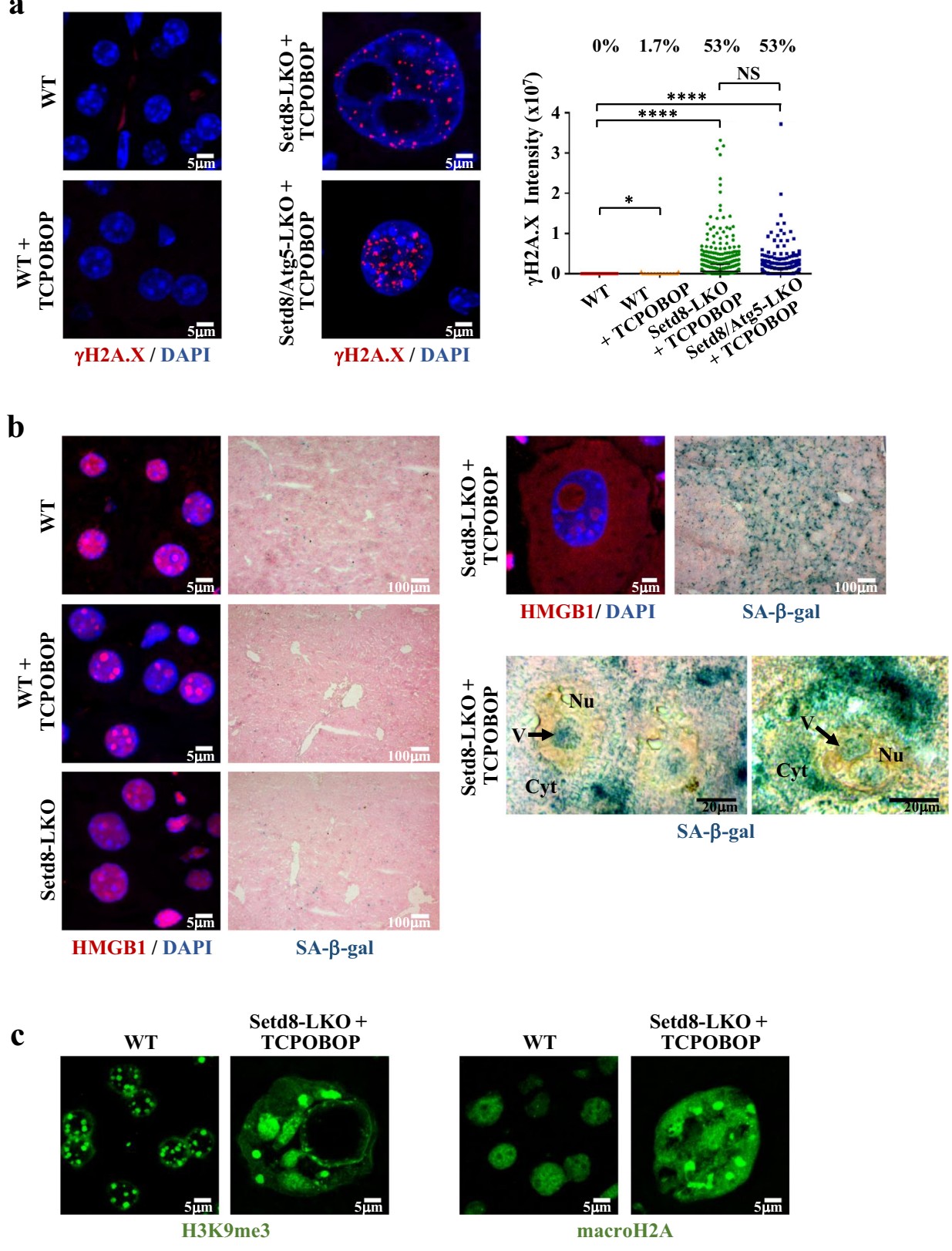

**Fig. 6 | Excessive DNA damage and senescence-related features in Setd8-LKO hepatocytes. a** Representative images of γH2A.X immunostaining of liver sections from wild type and TCPOBOP-treated wild type, Setd8-LKO and Setd8-LKO/Atg5-LKO double knockout mice. The graph shows HCM measurements of the γH2A.X immunofluorescence intensities in individual cells (*n* = 3412 for wild type; *n* = 3382 for Setd8-LKO + TCPOBOP; *n* = 3977 for WT + TCPOBOP and *n* = 3699 for Setd8-LKO/Atg5-LKO + TCPOBOP). Numbers indicate the percentage of cells with staining intensity above threshold. **b** Representative images of HMGB1 immunofluorescence and SA-β- galactosidase staining of liver sections 24 h after TCPOBOP treatment. Note the presence of SA-β-gal staining signal inside the vesicles in the close-up images of some Setd8-LKO hepatocytes. **c** Immunostaining with antibody recognizing trimethylated H3K9 and macroH2A. Note the appearance of large and diffuse H3K9Me₃-containing and macroH2A-containing heterochromatin foci in Setd8-LKO hepatocytes. Data analysis was performed using one-way ANOVA test. Data are presented as median values (black line) with SD. *p-value < 0.04; ****p-value < 0.0001. NS Non-significant. Source data are provided as a Source Data file.

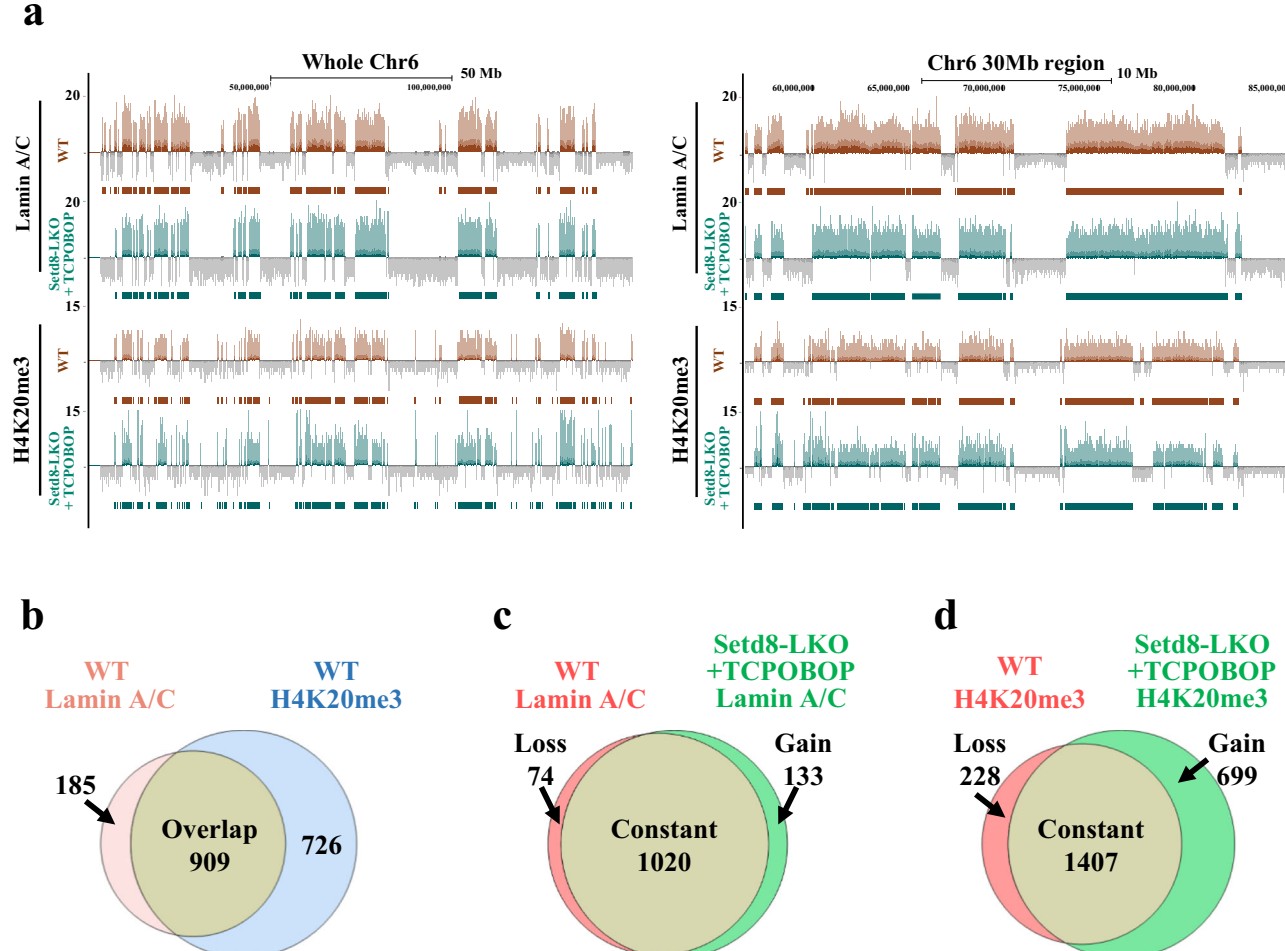

**Fig. 7 | Distribution of lamin-associated and H4K20Me3-modified genomic domains in Setd8-LKO hepatocytes. a** Genome Browser tracks showing normalized CUT&Tag reads of Lamin A/C-associated and H4K20Me3-occupied genomic regions on the entire chromosome 6 (left panel) and a 30 Mb region of chromosome 6 (panel at right). **b** Venn diagram showing the extent of overlap between Lamin A/C and H4K20Me3 CUT&Tag peaks in wild type liver nuclei. **c, d** Venn

diagram comparing Lamin A/C (**c**) and H4K20Me3 (**d**) CUT&Tag peaks between wild type and Setd8-LKO mice 24 h after TCPOBOP treatment. "Loss" indicates peaks present only in wild type livers, "Gain" indicates new peaks in Setd8-LKO livers, while "Constant" correspond to peaks found in both, the wild type and the Setd8-LKO livers.

## Discussion

Setd8 deficiency leads to genome instability, defects in mitotic chromatin compaction, cell cycle arrest[5–13] and metabolic reprogramming[16], which eventually culminate in necrotic cell death[15]. These effects are in apparent contradiction to the long-term survival phenotype and the late-onset development of liver cancer in liver-specific Setd8 knockout mice[15]. To reconcile this paradox, we analyzed the early stages of cell division-dependent hepatocyte death in Setd8-LKO mice.

We found that under mitogenic stress conditions, while most hepatocytes die via necrosis within 24 h, approximately 10% of the cells escape necrotic death by entering a state, which is characterized by several senescence features, including cell cycle withdrawal, DNA damage, senescence-related transcriptome changes, SA-β-galactosidase activity, SASP production, senescence-associated heterochromatin foci (SAHF) and nuclear enlargement. However, they did not lose lamin expression and also exhibited new features, such as transient hyperploidy and endonucleosis, not normally seen before in senescent cells.

Our results provide mechanistic explanations for the development of the above unusual phenotype, supporting a model schematically presented in (Fig. 10). The first step involves mitogenic stimulus-dependent endoreduplication in the cell division-incompetent, G2 phase-arrested Setd8-LKO hepatocytes. Endoreduplication generates

cells with multiple copies of each chromosome, which could confer a selective survival advantage by compensating for the detrimental effects of extensive DNA damage, such as transcriptional deregulation, the emergence of possible lethal loss-of-function mutations and/or haploinsufficiency.

Such genetic advantage, however, creates a physical situation where the "non-genetic" function of the genome becomes highly relevant: The cells, which escape necrosis and enter a senescent state, must cope with the increased nuclear DNA content either by increasing nuclear size or by initiating other structural changes in the nucleus. A plethora of previous studies have established the view that the shape of the nuclear envelope is subject to alterations regulated by internal forces connecting the lamin fibers to chromatin via LADs and external anchoring to microtubules and cytoskeletal filaments by nesprins and SUN domain proteins[40,41]. Setd8-deficient hepatocytes have greatly enlarged nuclei and consequently extended nuclear envelope, with increased elasticity. We observed thinner lamin fibers in Setd8-LKO cells and a proportionally increased lamin protein content per unit areas, which is indicative of a mechanical stretching of existing and new lamin polymers. The resulting elasticity of the nuclear envelope may not only allow for an increase in nuclear size, but may also counterbalance potential mechanical force-driven adverse effects caused by hyperploidy.

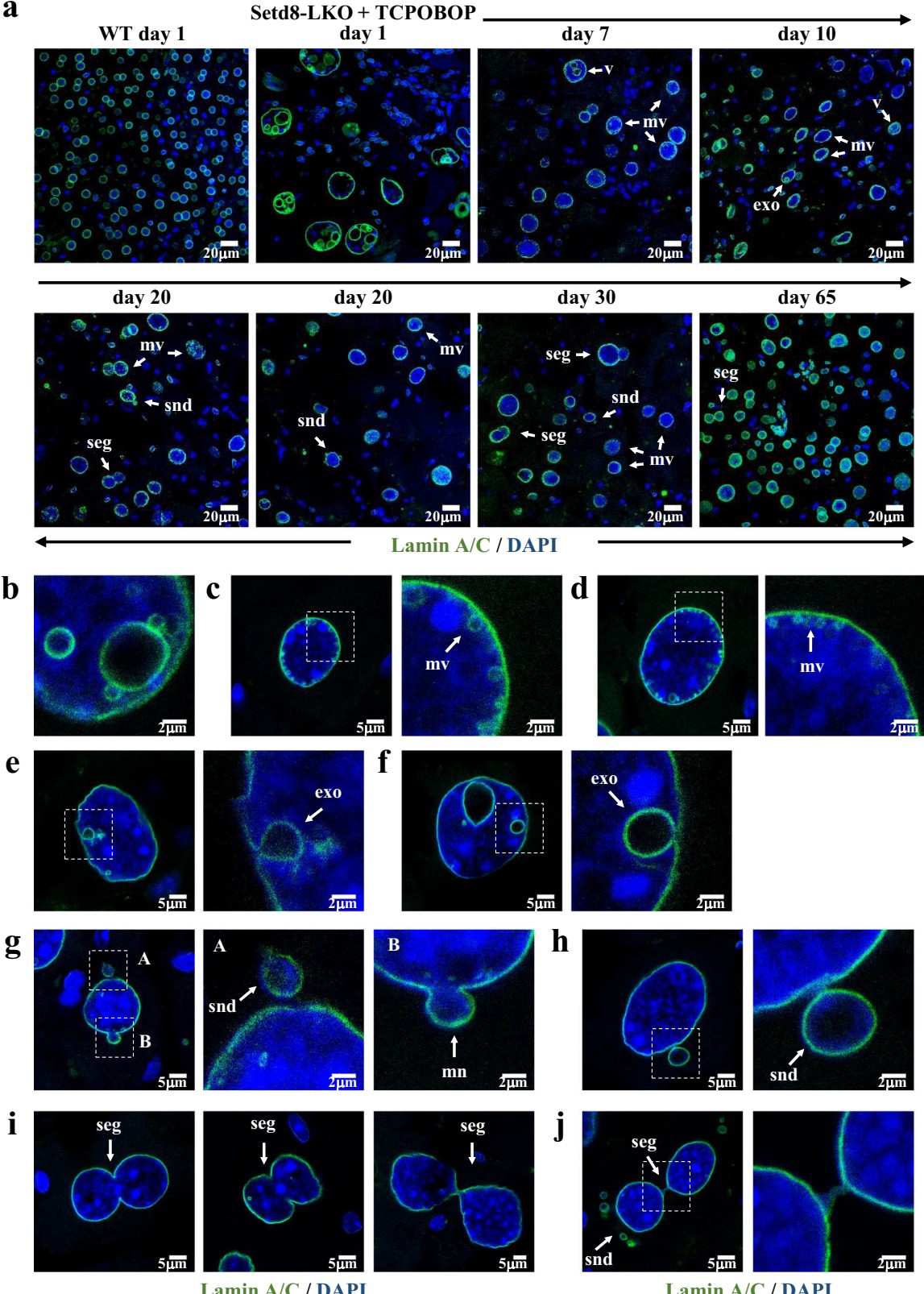

**Fig. 8 | Progressive loss of endonucleotic vesicles and segregation-mediated reduction of nuclear size at later periods. a** Representative images of Lamin A/C-stained hepatocyte nuclei in untreated wild type (WT day 1) and TCPOBOP-treated Setd8-LKO mice 1, 7, 10, 20, 30 and 65 days after the treatment. Arrows indicate: v, large vesicles; mv, microvesicles; exo, exonucleotic vesicles; snd, small nuclear detachments; seg, segregating nuclei. **b** Close-up image of separating microvesicles from a larger endonuleotic vesicle. **c**, **d** Microvesicles accumulating and fusing to the nuclear lamina. **e**, **f** Exonucleotic-like structures releasing vesicle content to the cytoplasm. **g**, **h** Nuclear blebbings and small nuclear detachments. **i** Segregating nuclei. **j** Segregating and budding nuclei.

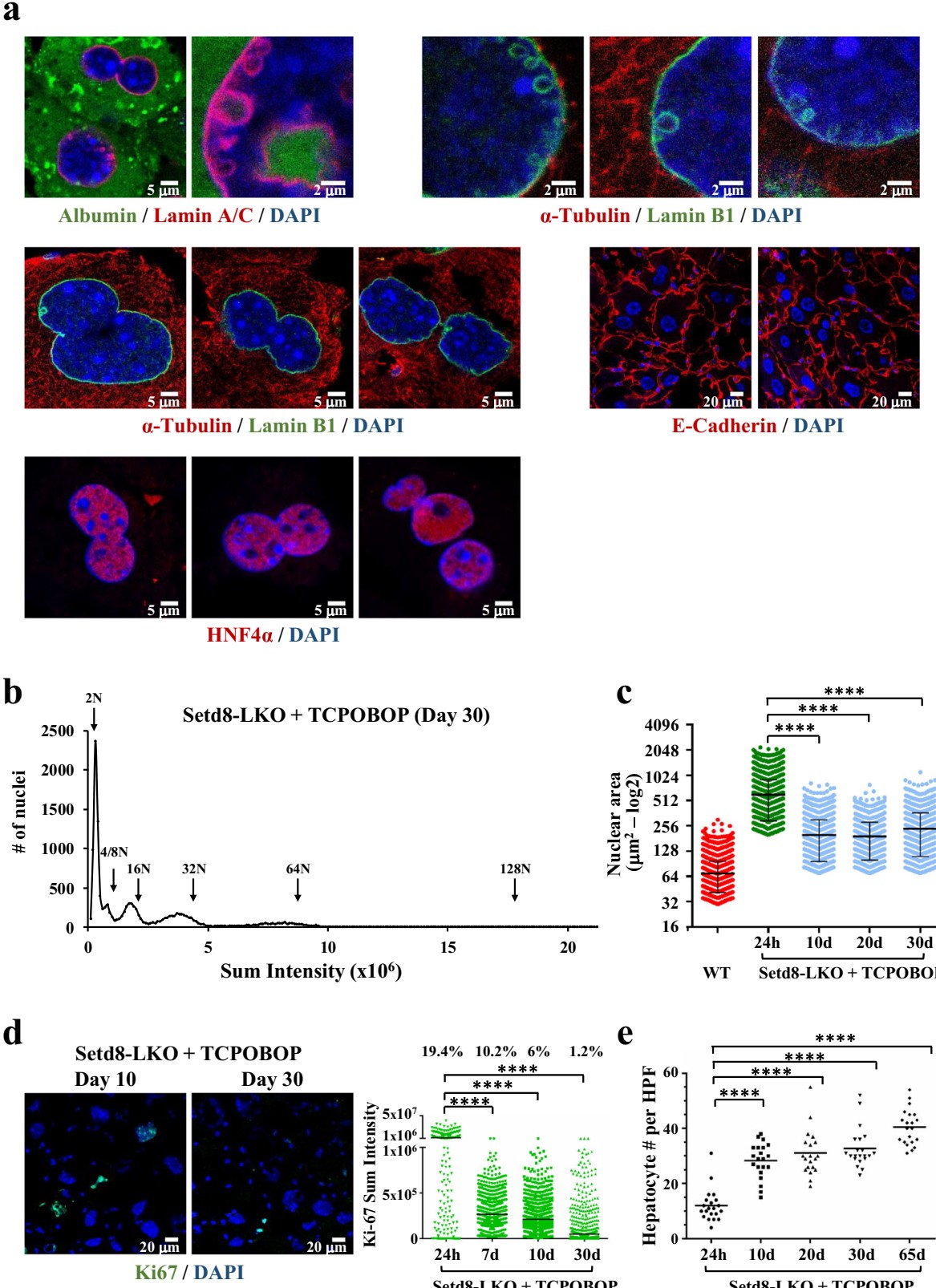

We also found that LADs are enriched with H4K20 trimethylated nucleosomes. Although Setd8-deficient cells cannot mono-methylate H4K20, the existing H4K20Me$_1$ histone tails can be up-methylated to di- and trimethylated forms by Suv4-20 h1/2[8,9]. Consistent with this, we observed highly elevated levels of H4K20Me$_3$ in Setd8-LKO mice, which can efficiently associate with the nuclear lamina and generate an additional internal force that folds the elastic nuclear envelope inwards.

Folding of the lamina towards the nuclear interior and the subsequent membrane closure generates intranuclear vesicles with entrapped cytoplasmic proteins, a process we term endonucleosis. Our results also demonstrate that the parallel activation and function

**Fig. 9 | Characteristics of Setd8-LKO livers 20–30 days after TCPOBOP treatment. a** Representative images of Setd8-LKO hepatocytes 20 days after TCPOBOP treatment stained for the cytoplasmic proteins Albumin and α-Tubulin, the nuclear protein HNF4α and the cytoplasmic membrane marker E-Cadherin. **b** Estimation of nuclear DNA content in isolated nuclei from Setd8-LKO livers ($n = 12569$) by propidium iodide (PI) staining 30 days after TCPOBOP treatment. Graphs show high-content microscopy (HCM) measurements of the PI staining intensity in individual cells. Arrows indicate the chromosomal ploidy of the peaks. The percentage of the cells corresponding to the peak areas were as follows: in 2 $N = 43.2\%$; -4/8 $N = 8.2\%$; -16 $N = 17.7\%$; -32 $N = 19.2\%$; -64 $N = 8\%$. Note the reappearance of cells with 2 $N$ DNA content and the substantial deviation of the peak centers from locations corresponding to sum intensities with doubling chromosome numbers, which is indicative of aneuploidy. **c** Comparison of the relative nuclear areas at different times following TCPOBOP treatment. The data for wild type and the 24 h time point of Setd8-LKO mice are the same as in Supplementary Fig. 1a, corresponding to the volumes of $n = 4295$ nuclei and $n = 4095$ nuclei, respectively. The data shown in

10 days, 20 days and 30 days after TCPOBOP treatment are measurements from $n = 3736$, $n = 3538$ and $n = 3690$ nuclei, respectively. **d** Representative images of liver sections stained for Ki67 in Setd8-LKO mice, 10 and 30 days after TCPOBOP treatment. The graph at right shows the quantification of the fluorescence sum intensities in cells stained positively for Ki67 at the time points 24 h ($n = 1228$), 7 days ($n = 1593$), 10 days ($n = 1534$) and 30 days ($n = 1495$) after TCPOBOP treatment. Note the gradual decrease of the percentage of cells with Ki67 signal above threshold and >5-fold decrease of the median Ki67 pixel intensity in the 7–10–30 days' time points. **e** Quantification of hepatocyte numbers in TCPOBOP-treated Setd8-LKO livers. The graph shows hepatocyte numbers in 20 randomly chosen High Power Fields (HPFs) from Setd8-LKO livers at the indicated time points after TCPOBOP treatment. Data analysis was performed using one-way ANOVA test. Data in (**c**, **d** and **e**) are presented as median values (black line). SD is indicated in (**c**). ****$p$-value < 0.0001. In (**e**) statistical comparisons were performed based on cell numbers in individual HPFs. Source data are provided as a Source Data file.

of the cytoplasmic autophagy machinery, which is induced by elevated AMPK levels in Setd8-LKO mice[16], is a prerequisite for facilitating and completing the endonucleosis process.

Given the context and the conditions under which this process takes place, it is tempting to speculate that endonucleosis represents an additional survival mechanism to counteract the extreme size expansion of the cells. Internalization of cytoplasm via autophagy could contribute to the elimination of toxic substances and structures that accumulate in the cytoplasm and to the preservation of the overall nucleus/cell size ratios within a range that is compatible with survival.

In line with this scenario, we found that the unusual phenotype characterized by hyperploidy and endonucleosis corresponds to an unstable and transient cellular state. A few days after their appearance, the endonucleotic vesicles became smaller and gradually disappeared. In parallel, the nuclear size and cellular ploidy were also reduced via mechanisms involving the segregation of the existing nuclei without mitosis and the extensive budding and detachment of smaller nuclear structures, reminiscent of micronuclei[37,38].

While these processes may act towards normalizing the physical characteristics of the surviving cells and their genome, they may also generate aneuploidy, which together with the activation of oncofetal genes and the parallel accumulation of inflammatory cells, could provide a fertile ground for the initiation of hepatocellular carcinoma at later stages.

Taken together, the senescence features described here, provide support for the concept of the non-genetic function of the genome and chromatin[42] by demonstrating their importance in the regulation of nuclear envelope plasticity in cellular survival under stress conditions.

## Methods

We confirm that the Ethical Review Board of FORTH has approved the study protocol. The research presented in this study complies with all relevant ethical regulations.

### Mice

*Setd8$^{lox/lox}$-AlbCre* (Setd8-LKO) mice were generated by crossing mice carrying floxed exon 7 allele[9] with Alb-Cre transgenic mice. Hepatocyte-specific, full inactivation of Setd8 in these mice was observed from postnatal day 20 and onwards[15]. *Ob/ob* mice carrying spontaneous mutation of the leptin gene (B6.Cg-Lep$^{ob}$/J, JAX stock 000632) were purchased from Jackson Laboratory. GFP-LC3 transgenic mice[27] (RBRC00806) and mice carrying floxed alleles of *Atg5* gene[43] (B6.129S-Atg5 [tm1Myok]) were from RIKEN BioResource Center and were obtained from Georgios Chamilos (Univ. of Crete). Setd8/Atg5-LKO double knockout mice were obtained by crossing *Setd8$^{lox/lox}$-AlbCre* mice with *Atg5$^{lox/lox}$* mice. The resulting single and double knockout mice were fertile and viable over 12 months of age. Mice were kept in

grouped cages in a temperature-controlled, pathogen-free facility on a 12 h light/dark cycle and fed with standard chow diet containing 19% protein and 5% fat (Altromin 1324) and water ad libitum. All animal experiments were approved by the Ethical Review Board of IMBB-FORTH and the Animal Ethics Committee of the Prefecture of Crete and were performed in accordance with the respective national and European Union regulations. All experiments were performed in randomly chosen age-matched male mice. No blinding was used in this study. Unless otherwise indicated, mice were treated and analyzed at 45–50 days after birth. Treatment of mice were performed with a single intraperitoneal injection of either vehicle (corn-oil) or 3 mg/kg 1,4-bis-[2-(3,5,-dichloropyridyloxy)] benzene (TCPOBOP, T1443, Sigma-Aldrich) or 2 g/kg Na-pyruvate (P2256, Sigma-Aldrich).

### Hematoxylin and Eosin (H&E) staining and immunofluorescence assays

Fixation, paraffin infiltration and H&E staining of liver tissues were performed as described previously[30]. For immunofluorescence staining, freshly isolated liver tissues were either prefixed in 4% paraformaldehyde (PFA) in PBS for 2 h and treated sequentially with 15% and 30% sucrose, before embedding in Optimal Cutting Temperature (OCT) embedding medium or embedded unfixed and were frozen in liquid nitrogen. Cryosections of pre-fixed liver tissue (5 – 7 μm thick) were boiled in 10 mM tri-sodium citrate buffer pH 6 with 0.05% Tween-20 twice for 5 min for heat-induced epitope retrieval, whereas frozen sections of unfixed tissue (5 – 7 μm thick) were air dried and fixed in 4% formaldehyde for 15 min at room temperature. The cryosections were blocked in 5% BSA in PBS containing 0.3% Triton X-100 for 1 h and then incubated with 1% BSA, 0.3% Triton X-100 in PBS with the indicated primary antibodies at 4$^0$C overnight. After incubation with AlexaFluor 594- or AlexaFluor 488-conjugated goat anti-rabbit or anti-mouse secondary antibodies for 1 h at room temperature and counterstained with 1 μg/ml DAPI for 10 min, the slides were covered with Mowiol® 4–88 Reagent (EMD Millipore, 475904). Fluorescence images were observed using a Leica TCS SP8 confocal microscope. Image analysis was performed with the FIJI/Image J software.

For chromosomal ploidy evaluation, isolated nuclei were seeded onto Superfrost® Plus glass microscope slides, fixed with 70% EtOH and stained with 50 μg/ml propidium iodide (Sigma-Aldrich, P4864) for 30 min. After washing with PBS and mounting with Mowiol® 4-88 Reagent, PI images were collected using Operetta High Content Screening Microscope (Perkin Elmer) and PI intensity data were evaluated using the Harmony Software 4.1 of PerkinElmer.

### High Content Microscopy analyses of wide field and confocal images

Tissue cryosections and isolated nuclei on slides labeled with immunofluorescence were captured with a 20x or 40x lens (Olympus–Shinjuku

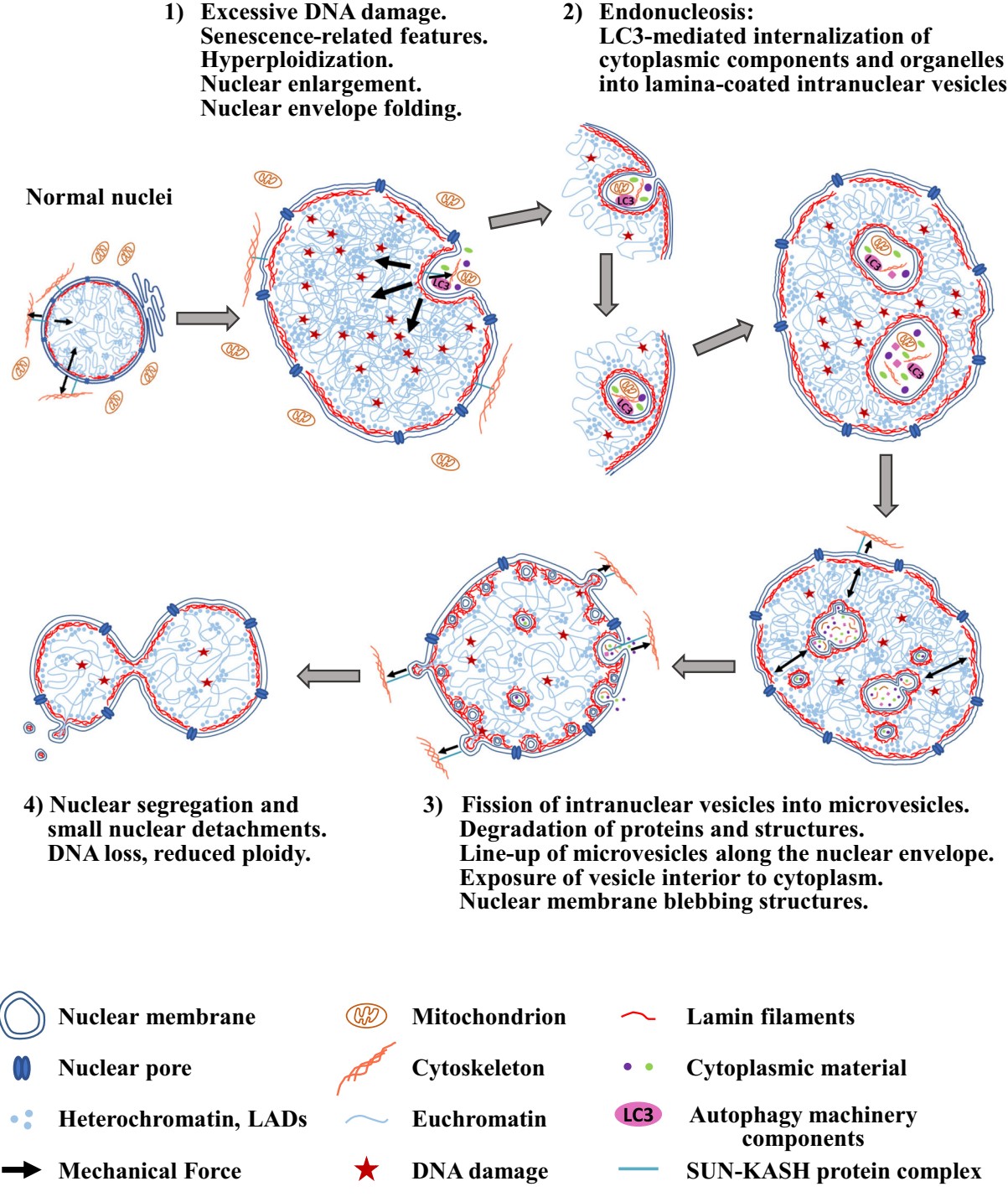

1) Excessive DNA damage.
   Senescence-related features.
   Hyperploidization.
   Nuclear enlargement.
   Nuclear envelope folding.

2) Endonucleosis:
   LC3-mediated internalization of
   cytoplasmic components and organelles
   into lamina-coated intranuclear vesicles.

4) Nuclear segregation and
   small nuclear detachments.
   DNA loss, reduced ploidy.

3) Fission of intranuclear vesicles into microvesicles.
   Degradation of proteins and structures.
   Line-up of microvesicles along the nuclear envelope.
   Exposure of vesicle interior to cytoplasm.
   Nuclear membrane blebbing structures.

Normal nuclei

| Symbol | Label | Symbol | Label | Symbol | Label |
|---|---|---|---|---|---|
| ◯ | Nuclear membrane | ⬭ | Mitochondrion | ∿ | Lamin filaments |
| ▮ | Nuclear pore | ⌇ | Cytoskeleton | • • | Cytoplasmic material |
| • : | Heterochromatin, LADs | ∼ | Euchromatin | LC3 | Autophagy machinery components |
| ➤ | Mechanical Force | ★ | DNA damage | — | SUN-KASH protein complex |

Fig. 10 | **Proposed model.** A schematic presentation of the dynamic nuclear features described in this study.

City, Tokyo, Japan) with the Operetta High Content Screening Microscope (HCSM) (PerkinElmer, Waltham, MA, USA) in wide field and analyzed using Harmony software 4.1 with PhenoLOGIC (PerkinElmer) or the open source software FIJI/Image J.

Segmentation of the nuclei was based on DAPI nuclear dye. Nuclei that were not intact, i.e. with no clear boundaries, or nuclei that were interrupted by the field boundaries were excluded from the analysis. Only whole, well-formed round nuclei were used for the analysis while non-hepatocyte cells were excluded based on their elongated shape and smaller size. For the measurement of nuclear size, the selected segmented nucleus area was measured in µm² in thirty different tissue sections and 5000 randomly selected nuclei were plotted in GraphPad

Prism 6. For the determination of the relative protein expression levels, the sum intensity of selected nuclei was calculated and plotted in GraphPad Prism 6. In the cases where background fluorescent noise was observed, this was estimated per nuclear area and subtracted from the nuclear sum intensity prior to plotting the graph. For the estimation of the number of nuclei per field in different time points, only fields with tissue coverage of 85 – 86% were selected.

For the construction of ploidy graphs, nuclei were captured with the 20x lens in the HCSM. Nuclei segmentation was based on the Propidium Iodide (PI) dye that quantifies the nuclear DNA content. Aggregates were excluded from the analysis. The PI sum intensity of each nucleus was calculated and was plotted in Microsoft Excel where

the $Y$ axis in the graphs represents the cell count within a range of sum intensities using a standard increase of 99999.

For the nuclear lamina intensity estimation, whole tissue cryosections were captured with a Leica TCS SP8 confocal microscope with a 63x lens and analyzed with using Fiji/Image J. One plane from the stack was selected, and the mean intensity of an area, delimited by tracing a two-pixel width line along the lamin-stained membrane over the point with the highest fluorescent intensity, was estimated. Measurements on a given nucleus were repeated multiple times and in multiple areas to minimize technical errors and the values were averaged per nuclei and plotted in Graphpad Prism 6. For estimation of nuclear lamina thickness, the function Plot Profile in FIJI/Image J was used, and the images were analyzed with the parameter setting 'full width at half-maximum', that gives the distance between the two intensity points in a selected area. For each nucleus, multiple measurements were performed in the same area. Areas that were not clear or the nuclear membrane was folded, were excluded from the measurement.

For the estimation of nuclear/cell area ratios, images from E-cadherin stained liver sections were evaluated by using FIJI/ImageJ analysis software. The perimeter of each cell and nucleus were marked by the Polygon selection tool. After clearing the background, nuclear and cellular areas were measured by the Analyze Particles tool using minimum area threshold of $50\,\mu m^2$ for wild type and $300\,\mu m^2$ for Setd8-LKO.

## Preparation of nuclei

Hepatocyte nuclei were prepared as follows: liver tissue was minced into small pieces in 20 volumes of ice-cold buffer containing 10 mM Tris pH 7.5, 2 mM $MgCl_2$, 3 mM $CaCl_2$ and protease inhibitor cocktail (Complete$^{TM}$ EDTA-free, Roche), and homogenized by 10 strokes using Dounce homogenizer. The suspension was filtered through $100\,\mu m$ mesh strainer and centrifuged for 5 min at 400 g, at $4^0$C. The resulting pellet was resuspended in a buffer containing 10 mM Tris pH 7.5, 2 mM $MgCl_2$, 3 mM $CaCl_2$, 10% glycerol, 1% NP40 and protease inhibitor cocktail (Complete$^{TM}$ EDTA-free, Roche). After incubation in ice for 3 min the nuclei were collected by centrifugation for 5 min at 400 g, at $4^0$C, washed twice with PBS and used for downstream applications.

## Electron microscopy

Liver tissues were fixed for 2 h at room temperature in 0.08 M sodium cacodylate buffer, containing 2% glutaraldehyde and 2% paraformaldehyde. After a 1 h post-fixation step with 1% osmium tetroxide, the slides were treated with 1% uranyl acetate for 20 min. The samples were dehydrated with serial ethanol treatment and subsequently embedded in Durcupan resin/propylene oxide (Polysciences). Approximately 100 nm thin sections on copper grids were observed at 80 kV with a JEOL JM2100 transmission electron microscope.

## 3-Dimensional DNA Fluorescence in situ hybridization (DNA FISH)

BAC clones were obtained from BACPAC Resource Center CHORI. The following BAC clones were used for probe preparation RP24-353H5 spanning *Afp* locus in chromosome 5; RP24-82E10 spanning *Pkm* locus in chromosome 9; RP24-389I13 for *Igf1* locus in chromosome 10.

BAC clones were labeled by nick-translation using Aminoallyl-dUTP-Cy3 (Jena Bioscience). 100 ng of the labeled probes were mixed with $1\,\mu g$ COT-1 DNA (Invitrogen), lyophilized, resuspended in formamide and incubated at $37^0$C for 30 min. After denaturation at $95^0$C for 10 min, the probes were diluted to prepare hybridization buffer containing 2xSSC, 10% Dextran Sulfate and 50 mM $Na_2HPO_4$.

$18\,\mu m$ thin liver cryosections were treated with 10 mM $Na_3$-citrate pH 6.0 for 10 min at $80^0$C and placed to room temperature for 1 h. The slides were sequentially treated with 70%, 80%, 95% and 100% ethanol for 3 min each. After DNA probe addition, the slides were incubated at $80^0$C for 5 min for additional denaturation, followed by hybridization at $37^0$C for 14 h. The slides were washed first with 0.5x SSC/0.1% Tween-20 for 5 min at $70^0$C. Next, the slides were washed with 1x SSC/0.1% Tween-20 for 5 min at room temperature, then with 4x SSC/0.1% Tween-20 for 5 min, by 2x SSC for 5 min and finally with PBS. The slides were further treated with 0.5% Triton X-100 in PBS for 24 h at $4^0$C, counterstained by DAPI and mounted using Prolong Gold Antifade reagent (Invitrogen).

## Senescence-associated beta-galactosidase (SA-β-gal) and PAS staining

Liver sections from OCT-embedded frozen livers were fixed in 0.2% glutaraldehyde for 10 min. The sections were washed with PBS and stained for 2 h with a solution containing 0.5 mg/ml X-Gal, 5 mM of $K_3Fe(CN)_6$, 5 mM $K_4Fe(CN)_6$, 2 mM $MgCl_2$, 150 mM NaCl, 40 mM citric acid/sodium dihydrogen phosphate pH 4. The sections were counterstained with eosin for 1.5 min and observed with light microscopy using Olympus CX23T microscope. Images were taken using Axiocam ERc5s (ZEISS) and the ZEN microscopy software (ZEISS).

Periodic-acid Schiff (PAS) staining was performed in formalin-fixed paraffin-embedded liver sections. The sections were treated with 0.5% Periodic acid for 5 min and stained in Schiff's reagent (Merck) for 10 min. After washings with PBS, the sections were counterstained with hematoxylin and observed by light microscopy as above.

## Proximity Ligation Assay (PLA)

Proximity Ligation Assays were performed in liver cryosections from frozen tissue using Duolink® In situ Red Starter Kit Mouse/Rabbit from Sigma-Aldrich, according to the manufacturer's instructions. The pairs of primary antibodies were mouse anti-LaminA/C (Cell Signaling Technology, #4777), with rabbit anti-LaminB1 (Abcam, ab16048); and mouse anti-LaminA/C (Cell Signaling Technology, #4777) with rabbit anti-H4K20Me$_3$ (Abcam, ab9053). Following PLA reaction, the slides were counterstained with $1\,\mu g/ml$ DAPI for 10 min and observed in Leica TCS SP8 confocal microscope.

## CUT&Tag and ATAC-seq assays

CUT&Tag assays were performed as follows: $10^5$ nuclei were resuspended in a buffer containing 20 mM HEPES-KOH pH 7.9, 10 mM KCl, 0.1% Triton X-100, 20% Glycerol, 0.5 mM Spermidine, protease inhibitor cocktail (Complete$^{TM}$ EDTA-free, Roche) and incubated for 10 min with Concanavalin A-conjugated magnetic beads (Epicypher) that were previously activated by repeated washings with a buffer containing 20 mM HEPES-KOH pH 7.9, 10 mM KCl, 1 mM $CaCl_2$, 1 mM $MnCl_2$. The reactions were supplemented with 0.5 volume of a buffer containing 20 mM HEPES-NaOH pH 7.5, 150 mM NaCl, 0.5 mM Spermidine, protease inhibitor cocktail, 0.01% Digitonin, 2 mM EDTA and $0.5\,\mu g$ of the anti-Lamin A/C or anti-H4K20Me$_3$ or IgG negative control primary antibodies. The samples were incubated overnight at $4^0$C by gentle agitation. The beads were collected and supplemented with a buffer containing 20 mM HEPES-NaOH pH 7.5, 150 mM NaCl, 0.5 mM Spermidine, protease inhibitor cocktail, 0.01% Digitonin and $0.5\,\mu g$ of anti-rabbit or anti-mouse secondary antibodies (Epicypher). After incubation for 30 min at room temperature, the beads were washed twice with the same buffer and resuspended in the same buffer containing 300 mM NaCl. Protein A/G-fused Tn5 transposase protein (CUTANA pA/G-Tn5, Epicypher) was added and the samples were incubated for 1 h at room temperature followed by washing with a buffer containing 20 mM HEPES-NaOH pH 7.5, 300 mM NaCl, 0.5 mM Spermidine, protease inhibitor cocktail and 0.01% Digitonin. The beads were collected and resuspended in the above buffer containing 10 mM $MgCl_2$, and incubated for 1 h at room temperature. The beads were collected and washed once with 10 mM TAPS pH 8.5 and 0.2 mM EDTA. DNA was released from the beads by incubation in 10 mM TAPS pH 8.5 and 0.1% SDS for 1 h at $58^0$C. After quenching by the addition of

3 volumes of 0.67% Triton-X100, the samples were used for PCR reaction (15 cycles) using universal i5 primers and barcoded i7 primers. After cleanup with AMPure beads (Beckman Coulter), quantification and quality evaluation in Agilent Bionalyzer, the libraries were sequenced using Illumina NextSeq500 platform.

For ATAC-seq reactions we used the ATAC-seq kit from Active Motif. Tagmentation reactions using $10^5$ nuclei, DNA purification and PCR amplifications using combinations of indexed i5 and i7 primers were performed according to the manufacturer's instructions. The resulting libraries were sequenced using Illumina NextSeq500 platform.

### Chromatin Immunoprecipitation (ChIP) assays

ChIP assays were performed as described previously[31,44]. Briefly, livers were minced to small pieces and crosslinked with 1% formaldehyde for 10 min. After the addition of glycine at 0.125 M final concentration, the crosslinked cells were washed with a buffer containing 50 mM HEPES pH 7.9, 100 mM NaCl, 1 mM EDTA, 0.5 mM EGTA and sequentially treated with a buffer containing 0.25% Triton-X100, 10 mM EDTA, 0.5 mM EGTA, 20 mM HEPES pH 7.9 and a buffer containing 0.15 M NaCl, 1 mM EDTA, 0.5 mM EGTA, 20 mM HEPES pH 7.9 for 10 min each. The resulting crosslinked nuclei were resuspended in 10 volumes of 50 mM HEPES pH 7.9, 140 mM NaCl, 1 mM EDTA, 0.1% Na-deoxycholate, 0.5% Sarkosyl, and protease inhibitor cocktail (Complete™ Roche) and sonicated for 10 min in Covaris Sonicator instrument with maximum setting. After sonication, the samples were supplemented with 0.5 volume of 50 mM HEPES pH 7.9, 140 mM NaCl, 1 mM EDTA, 3% Triton X-100, 0.1% Na-deoxycholate, and protease inhibitor cocktail, and centrifuged for 15 min at 20.000 g. The soluble chromatin was incubated overnight with Dynabeads Protein G (Invitrogen), that were prebound by 5 μg of the primary antibodies recognizing H3K27ac-modified histones, CTCF and Smc3. The beads were washed twice with a buffer containing 50 mM HEPES pH 7.9, 140 mM NaCl, 1 mM EDTA, 1% Triton X-100, 0.1% Na-deoxycholate, 0.1% SDS, 0.5 mM PMSF, 2 μg/ml aprotinin, once with the same buffer but containing 500 mM NaCl, once with a buffer containing 20 mM Tris, pH 8.0, 1 mM EDTA, 250 mM LiCl, 0.5 % NP-40, 0.5 % Na-deoxycholate, 0.5 mM PMSF, 2 μg/ml aprotinin and once with a buffer containing 10 mM Tris pH 8.0, 0.1 mM EDTA, 0.5 mM PMSF, 2 μg/ml aprotinin. Immunoprecipitated chromatin was eluted from the beads and reverse crosslinked by overnight incubation in a buffer containing 50 mM Tris-HCl pH 8.0, 1 mM EDTA, 1% SDS and 50 mM NaHCO$_3$ at 65$^0$C. The samples were then incubated with 25 μg/ml RNase-A for 30 min at 37$^0$C, followed by incubation with 50 μg/ml Proteinase-K for 2 h at 42$^0$C. DNA was extracted by phenol/chloroform and precipitated with ethanol.

Libraries were prepared from 10 ng input DNA NEBNext Ultra II DNA Library Prep *Kit* of New England Biolabs according to the manufacturer's instructions and sequenced in Illumina NextSeq500 sequencer.

### CUT&Tag, ATAC-seq and ChIP-seq data analysis

Computational analyses of genome-wide mapping data were performed as described previously[30,44], with minor modifications. Briefly, before performing any alignment or additional analysis, the quality of all the FASTQ files and the reads they contain was assessed using *FastQC* https://www.bioinformatics.babraham.ac.uk/projects/fastqc. To remove low-quality bases and adapter sequences from the raw FASTQ files, we employed *Trimmomatic* (version 0.39)[45]. Duplicate reads were identified and removed using *samtools* version 1.10[46]. The FASTQ files were mapped on the UCSC mm10 genome using *hisat2* with default settings[47]. We retained only the reads with a mapping quality score >30 and removed blacklisted regions from our BAM files which are known to be susceptible to technical artifacts and spurious signals. The resulting BAM files were converted to bigwig files using *deeptools* (version 3.3.2)[48] and visualized in the UCSC Genome Browser.

Before identifying peak regions and to account for variations in the number of sequencing reads between ChIP and CUT&Tag samples, we standardized the total read count for each sample by equally reducing reads proportionally based on the sample with the fewer reads. Peaks were called using *MACS2* (2.2.7.1) for mouse genome with a *p value cutoff* of 1.00e-13(2.2.7.1)[49]. In the case of Lamin A/C and H4K20Me$_3$ CUT&Tag data, peak calling was performed by SICER2[50].

De novo motif search was run using HOMER45 within the median +/- nt intervals of each sample separately around the peak summit of 25% best scoring (pvalue) peaks.

To gain insightful visual representations of the genomic binding patterns and chromatin profiles, we employed *deeptools* (version 3.3.2) utilizing the "computeMatrix" module and calculated the read density at the peak centered regions. Venn diagrams of overlapping regions between conditions were created using custom made scripts in R version (4.3.0).

### RNA purification and RNA-sequencing

Total RNAs were purified from liver tissues by brief homogenization in 10 volumes of Trizol reagent using Polytron device, followed by incubation at room temperature for 5 min, the addition of 0.2 volumes of chloroform and further incubation at room temperature for 3 min. The samples were centrifuged at 12000 g for 15 min at 4$^0$C, the aqueous phase was collected and was precipitated by the addition of equal volume of isopropanol. After 10 min at room temperature, the RNA was collected by centrifugation at 12000 g for 15 min. The pellet was resuspended in H$_2$O and re-precipitated with ethanol. The RNA samples were further purified by digestion with 10 units of DNase-I for 10 min at 37$^0$C, followed by purification with phenol/chloroform extraction and ethanol precipitation.

RNA-seq libraries were generated using NEBNext Ultra II RNA Library preparation kit from New England Biolabs and sequenced in an Illumina NextSeq 500 system. Raw sequence data were quality-assessed and pre-processed using PRINSEQ version 0.20.4[51]. Pre-processed FASTQ files were subsequently mapped to the UCSC mm10 reference genome, using HISAT2 version 2.1.0 with the argument –score-min L,0.0–0.5, setting the function for the minimum valid alignment score. The resulting BAM files were analyzed with the Bioconductor package metaseqR[52]. Differential gene expression analysis and visualizations were performed as described previously[30]. Gene Set Enrichment Analysis (GSEA) were performed using the GSEA software version 3.0[53], with default parameters and FDR < 0.25 cutoff. The analysis was performed on the RPGM normalized expression values of the differentially expressed genes.

### Quantification and statistical analysis

Comparisons between two groups were performed using Student unpaired *t*-test or one-way ANOVA. All statistical analyses were performed with Graphpad Prism 6 to 8 (GraphPad).

Sample size was determined empirically based on similar studies with Setd8-LKO mice (Refs. 15,16). No statistical method was used to predetermine sample size. As a rule, all of the imaging-based assays were performed at least 2 different dates with 3 different biological replicates (mice) each time and at least 8–0 technical replicates (sections) each time. RNA-seq analyses involved RNAs from 4 and 5 biological replicates collected at the same date. ChIP-seq and CUT&Tag assays involved at least 2 biological replicates.

### Reporting summary

Further information on research design is available in the Nature Portfolio Reporting Summary linked to this article.

## Data availability

The High Content Microscopy data generated in this study are provided in the Source Data file. RNA-seq, ChIP-seq, ATAC-seq and CUT&Tag data have been deposited in the Gene Expression Omnibus (GEO) under accession number GSE242719. Source data are provided with this paper.

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

## Acknowledgements

We thank H. Kontaki, M. Koukaki and E. Moltsanidou for technical assistance and discussions during the course of the work; E. Stratidaki, N. Gounalaki and E. Dialynas of the IMBB-Genomics facility for assistance with NGS library preparations and sequencing; S. Papadogiorgaki for help with electron microscopy; M. Vasilarou for initial RNA data analyses; P. Hatzis for editing the manuscript. This work was supported by the European Union, ERC Advanced Investigator Grant (ERC-2011-AdG294464), the "Basic research Financing (Horizontal support of all Sciences)" under the National Recovery and Resilience Plan "Greece 2.0" funded by the European Union – NextGenerationEU, H.F.R.I. Project:15276, and the AXA Research Fund Chair in Epigenetics Program #2016 (to I.T.) and pre-doctoral fellowships from the Hellenic Foundation for Research and Innovation #10442 and #10444 (to O.G and E.T.).

## Author contributions

K.C.N. and I.T. conceptualized and initiated the project. O.G. and I.T designed the study, performed experiments and evaluated the data. E.T., E.D., S.G. performed experiments and analyzed the data. D.B. performed bioinformatics analyses. Y.D. and G.C. supervised the Electron Microscopy experiments. I.T. supervised the study and wrote the manuscript. All of the authors reviewed and edited the manuscript.

## Competing interests
The authors declare no competing interest.
