## [Peer Review File · Nature Communications]

Endonucleosis mediates internalization of cytoplasm into the nucleusREVIEWER COMMENTS

Reviewer #1 (Remarks to the Author):

Galanopoulou et al. describe an interesting phenomenon, i.e. that in mitogen-stimulated liver-specific Setd8-KO mice senescent, not necrotic hepatocytes show nuclear engulfments progressing to the formation of intranuclear vesicles, a process that they name "endonucleosis". They suggest that this phenomenon is an expression of a survival mechanism to prevent necrotic death.

Overall, the paper is well written and, in particular, the images presented well detail the occurrence of these phenomena at the nuclear envelope level.

Although this Reviewer finds it highly likely that the observed alterations of the nuclear envelope involve both inner and outer nuclear membrane, the authors stain the nuclear envelope with SUN2 only; they need a marker of the outer nuclear membrane, like VAP-A to show that the nuclear envelope invaginations are of type II, not type I and the process of "endonucleosis" involves the outer nuclear membrane as well.

Also, the authors should quantitate the number or proportion of cells displaying clear signs of "endonucleosis" and of cells that recover after "endonucleosis", otherwise it is possible to imagine that endonucleosis and the full recovery are sporadic events.

Reviewer #2 (Remarks to the Author):

This manuscript reports endonucleosis in mitogen-stimulated liver-specific Setd8-KO mice. The authors claim that the cells undergoing endonucleosis are senescent, proposing that this is a unique feature of senescence phenotype, functioning as part of survival mechanisms to prevent necrosis.

Major issues are the artificial condition used to create endonucleosis and the premature definition of senescence.

1. Endonucleosis is only seen in liver-specific Setd8 knockout mice (Setd8-LKO) with TCPOBOP, a CAR agonist. These conditions are not seen in usual contexts associated with senescence induction and hence are highly artificial. The authors do not report whether endonucleosis is seen in other contexts of senescence or pathological conditions. It is difficult at this stage to justify the significance of the finding.

2. The definition of "senescence" are problematic.

- The evidence supporting a senescent state includes SA-beta-Gal activity and the nuclear exit of HMGB1. The image of SA-beta-Gal is low-magnification, and I cannot tell whether the positive cells are from the hepatocytes. In fact, macrophages tend to stain high in SA-beta-Gal. KO Setd8 affects chromatin, questioning whether the change of HMGB1 is a consequence of Setd8 or a marker of senescence.

- Several evidence argues against that these cells are senescent: (1) These cells actively incorporate Ki67, which is not seen in traditionally defined senescent cells. (2) These cells maintain Lamin B1, the loss of which is a hallmark of senescent cells. (3) These cells increase in overall numbers, contradictory to the common notion that senescent cells are a stable form of cell cycle arrest.

Minor:

1. Diagram mentioned "micronuclei". Micronuclei specifically happens during mitosis, referring to the chromosome segregation error. Here the cells are not dividing and the structures are therefore not micronuclei.

2. MAP1LC3a mice are MAP1LC3B. LC3-GFP should be termed GFP-LC3.

Reviewer #3 (Remarks to the Author):

In liver-specific Setd8-KO mice treated with TCPOBOP, Galanopoulou et al. have identified a unique senescence-like phenotype in hepatocytes, accompanied by hyperploidy and formation of intranuclear vesicles (which they termed "endonucleosis"). The process of endonucleosis appears to be dependent on intact autophagy. The phenotype is quite striking and novel, but the data are largely descriptive. It is unclear what the biological significance of the phenotype is and how it can be generalized.

The Atg5-dependency of the phenotype is interesting, providing some mechanistic insight, but the analysis is limited. E.g. in Sup. Fig. 1c, the increased nuclear area in setd8-LKO+TCPOBOP is partially reverted with Atg5 KO. Is this due to reduced intranuclear vesicle formation (i.e. hyperploidization still occurs), or is endoreduplication also inhibited with Atg5 deficiency? It would be important to clarify which aspects of the setd8-LKO+TCPOBOP phenotype are Atg5-dependent/Atg5-independent.

Autophagy is typically cytoprotective but could also trigger cell death, particularly if its activity is exceptionally high. I wonder whether the massive cell death observed in the setd8-LKO+TCPOBOP liver is 'autophagic cell death'.

"At 8 to 12 months of age, Setd8-deficient mice develop full-blown spontaneous liver tumors". What about setd8-LKO+TCPOBOP mice - are they more tumorigenic?

Atg5 deficiency can be tumorigenic in the liver (e.g. PMID: 21498569). Do the TCPOBOP-treated Setd8/Atg5 double knockout mice show any acceleration of tumor development?

I wonder if they can identify a similar phenotype in different contexts: how about e.g. setd8 ko fibroblasts with mitogenic stress in culture?

Responses to Reviewers' Comments

We would like to thank the Reviewers for the in-depth reviews and the constructive criticism on some key issues that required further attention and clarifications, and for the specific suggestions that helped us to prepare an improved version of the paper. We have addressed each point by providing new data and also by text corrections or better explanations.

The new data in the Revised version include:

- 1) Additional control HMGB1 and SA- β -gal staining in untreated Setd8-LKO liver sections (Fig. 4b).
- 2) Close-up pictures of SA- β -gal staining in TCPOBOP-treated Setd8-LKO livers, showing SA- β -gal in the intranuclear vesicles (Fig. 4b).
- 3) Immunostaining with macroH2A antibody, an independent senescence marker (Fig. 4c).
- 4) Immunostaining with an antibody detecting Nup98, a nuclear pore complex component located in both inner and outer nuclear membrane. (Supplementary Fig. 2d).
- 5) Detection of endonucleosis in Setd8-LKO livers under different metabolic stress conditions:
 - a) Fasting (Supplementary Fig. 4a).
 - b) Na-Pyruvate challenge (Supplementary Fig. 4b).
- 6) Detection of endonucleosis in ob/ob mice by
 - a) The detection of glycogen in intranuclear vesicles (Supplementary Fig. 4c)
 - b) The detection of intranuclear vesicles by Lamin A/C staining (Supplementary Fig. 4d)
- 7) Detection of γ H2AX staining in endonucleotic vesicle-containing ob/ob hepatocytes (Supplementary Fig. 4e).
- 8) Detection of H4K20Me₁ staining in endonucleotic vesicle containing ob/ob hepatocytes (Supplementary Fig. 4e).
- 9) Estimation of DNA content in the nuclei of Setd8/Atg5 double KO hepatocytes by propidium iodide staining (Supplementary Fig. 5c).
- 10) DNA FISH assay data in the nuclei of Setd8/Atg5 double KO hepatocytes (Supplementary Fig. 5d).

Point-by-point responses to Reviewers' comments:

Reviewer 1. Point 1

Although this Reviewer finds it highly likely that the observed alterations of the nuclear envelope involve both inner and outer nuclear membrane, the authors stain the nuclear envelope with SUN2 only; they need a marker of the outer nuclear membrane, like VAP-A to show that the nuclear envelope invaginations are of type II, not type I and the process of "endonucleosis" involves the outer nuclear membrane as well.

Response: In the revised Supplementary Fig. 2d we included immunostaining with Nup98 antibody. The data show that Nup98, a nuclear pore complex component located in both inner and outer nuclear

membrane, can be readily detected in the intranuclear vesicle membranes, which suggests that the nuclear envelope invaginations involve both inner and outer nuclear membrane.

We have also stained liver tissue sections with two different VAP-A antibodies (one from Proteintech #15275-1-AP and another one from Santa Cruz Biotechnology sc-293278). We expected staining the cytoplasmic membranes with some enrichment around the nuclear membrane. As shown below in representative focal plane pictures, VAP-A signal was detected in the cytoplasm of WT hepatocytes without an enrichment around the nuclear membrane. In Setd8-LKO+TCPOBOP samples, VAP-A was detected in the intranuclear vesicle interior. Although this pattern is compatible with our conclusions, the relatively uniform cytoplasmic signal in WT cells, raises some uncertainty about the specificity of the observed signal. Therefore, we prefer to present only the evidence provided by Nup98 detection in the Revised version.

Reviewer 1. Point 2

Also, the authors should quantitate the number or proportion of cells displaying clear signs of “endonucleosis” and of cells that recover after “endonucleosis”, otherwise it is possible to imagine that endonucleosis and the full recovery are sporadic events.

Response: After examining max-projections of more than one hundred High Power Fields manually, we could not detect cells without signs of endonucleosis in TCPOBOP-treated Setd8-LKO mice at 24 hours. Even in the few cases where the enlarged cells lacked clearly evident Lamin A/C-coated vesicles, we could observe at least one vesicle in specific focal planes. Similarly, we have examined sections at day 65. In hundred different fields we found one or two cells which retained intranuclear vesicles. We did not perform HCM quantification of the cells, since it was not possible to use settings that can distinguish between vesicle-containing and not containing cells. Therefore, we indicate in page 2, that “The vast majority, essentially all (>99%) of the enlarged nuclei contained Lamin A/C and Lamin B1-coated heterogeneously-sized vesicles”; and in page 6, that “Thirty to 65-days after treatment, intranuclear vesicles were essentially absent (i.e. observed less than 1 cell with vesicles per 50 fields)...”

Reviewer 2. Point 1

Major issues are the artificial condition used to create endonucleosis and the premature definition of senescence.

1. Endonucleosis is only seen in liver-specific Setd8 knockout mice (Setd8-LKO) with TCPOBOP, a CAR agonist. These conditions are not seen in usual contexts associated with senescence induction and hence are highly artificial. The authors do not report whether endonucleosis is seen in other contexts of senescence or pathological conditions. It is difficult at this stage to justify the significance of the finding.

Response: We agree with the note that the importance of the endonucleosis process would be strengthened if we can demonstrate it in other physiological or pathological conditions besides the TCPOBOP-induced replication stress condition. In this regard, we note that we have previously shown that in Setd8-LKO livers contain enlarged hepatocytes, initially (at 3-months of age) detectable in small areas, where only few cells physiologically enter the cell-cycle (Ref. 15). These areas gradually expand over time, because Setd8-deficient cells cannot progress beyond the next G2-phase and the resulting cell death triggers a regenerative process, whereby neighboring cells are forced to enter the cell cycle, resulting in a futile cycle of cell death and replenishment. In Ref 15, we also showed that all of the Setd8-LKO hepatocytes become larger, after 2/3rd partial hepatectomy, a condition when all of the hepatocytes (which were normally in G0 phase), synchronously enter into a proliferation state. These results suggested that hepatocyte and nuclear enlargement is dependent on proliferation.

In the current paper we induced proliferation by mitogen (TCPOBOP) treatment, to achieve homogenous cellular profiles that would allow reliable quantitation of cellular changes and the execution of genomic studies.

In Ref. 16, we examined Setd8-LKO livers in an earlier age (45-days after birth) and showed that the apparently normal hepatocytes, undergo a dramatic metabolic reprogramming and parallel activation of autophagy. These cells are highly sensitized to different stress conditions.

In the original version of the paper we showed that enlarged Setd8-LKO hepatocytes display endonucleosis and hyperploidization **under proliferation condition**, induced by TCPOBOP.

In the revised version we provide evidence that endonucleosis is also triggered by two different metabolic stress conditions, such as **fasting** and **Na-pyruvate challenge** (Supplementary Fig. 4a and 4b).

In addition, we could also detect **spontaneously occurring** (i.e. without any treatment) endonucleosis in about **10-15% of the hepatocytes of ob/ob mice** (Supplementary Fig. 4c and 4d). The leptin-deficient ob/ob mice, fed with standard chow diet, accumulate triglycerides in hepatocytes, which imposes a different type of metabolic stress.

Importantly, the **ob/ob model provides an example for the occurrence of endonucleosis independently of Setd8 function**. This is supported by the normal H4K20Me₁ pattern in cells displaying endonucleosis (Supplementary Fig. 4e). Interestingly, in ob/ob mice, the cells containing endonucleotic vesicles were selectively stained positively for γ H2A.X (Supplementary Fig. 4e), suggesting that DNA damage could be an important condition that triggers nuclear membrane processes leading to endonucleosis. This correlation further strengthens our interpretation that some cells with high accumulation of genome damage may escape cell death by endoreduplication-mediated hyperploidy, which buffers-out possible lethal mutations, and that the increased nuclear DNA content could initiate structural changes in the nucleus that eventually lead to endonucleosis.

Apart from the different stress conditions presented in the Revised version i.e a) mitogenic stimulus by TCPOBOP; b) metabolic stress by fasting; c) metabolic stress by pyruvate challenge; and d) spontaneous occurrence in ob/ob mice accumulating lipids in hepatocytes, we have noticed a similar phenomenon in a recent paper analyzing Arid1a-deficient mice.

In the March 15 issue of Science Advances, D'Ambrosio et al. show that *Arid1a*-deficiency leads to genomic instability triggering the development of aggressive tumors (we mention it in Ref. 25 in the Revised paper). Although the study focuses on the genomic features of the process, they also detect some nuclear abnormalities, which resemble to endonucleosis, along with micronuclei formation. The authors provide an H&E staining picture with cells containing vesicles but do not further characterize them. To help the Reviewers, we copy here the respective data from Fig S7:

Figure S7 B and C from the paper: D'Ambrosio A, Bressan D, Ferracci E, Carbone F, Mulè P, Rossi F, Barbieri C, Sorrenti E, Fiaccadori G, Detone T, Vezzoli E, Bianchi S, Sartori C, Corso S, Fukuda A, Bertalot G, Falqui A, Barbareschi M, Romanel A, Pasini D, Chiacchiera F. Increased genomic instability and reshaping of tissue microenvironment underlie oncogenic properties of *Arid1a* mutations. *Sci Adv.* 2024 Mar 15;10(11):eadh4435. doi: 10.1126/sciadv.adh4435.

We think that the cells shown in the H&E section clearly correspond to membrane-coated intranuclear vesicles-containing cells and the Lamin A/C staining figure shows enlarged nuclei and several micronuclei. Although the cells shown in Lamin A/C panel do not seem to have intranuclear vesicles, we assume that this profile corresponds to a later stage of DNA damage induced nuclear alterations, when the vesicles are gradually eliminated. The results of this study provide an additional example for the occurrence of endonucleosis independently of *Setd8* function.

Taken together, the above results suggest that endonucleosis operates in a variety of conditions. We note that at least in the above mentioned physiological and pathological contexts, DNA damage accumulation is a feature that coincides in cells displaying endonucleosis.

We also speculate that in many cases, due to the fast turnover of the intranuclear vesicles, the process often escapes detection in more other contexts, which are accompanied by excessive DNA damage.

Reviewer 2. Point 2

The definition of “senescence” are problematic.

*- The evidence supporting a senescent state includes SA-beta-Gal activity and the nuclear exit of HMGB1. The image of SA-beta-Gal is low-magnification, and I cannot tell whether the positive cells are from the hepatocytes. In fact, macrophages tend to stain high in SA-beta-Gal. KO *Setd8* affects chromatin, questioning whether the change of HMGB1 is a consequence of *Setd8* or a marker of senescence.*

- Several evidence argues against that these cells are senescent: (1) These cells actively incorporate Ki67, which is not seen in traditionally defined senescent cells. (2) These cells maintain Lamin B1, the

loss of which is a hallmark of senescent cells. (3) These cells increase in overall numbers, contradictory to the common notion that senescent cells are a stable form of cell cycle arrest.

Response: We agree with the Reviewer and also support the policy that the use of the term “senescent cell”, should be restricted only to the cases displaying all of the major hallmark features. Even though many senescent features are detectable in Setd8-LKO hepatocytes, because of the maintenance of Lamin B1 expression, whose loss is an important hallmark of senescent cells, the use of the term senescent state could be misleading.

Therefore, we have made the necessary corrections in the title, abstract and throughout the text and avoid calling the surviving cells senescent.

In describing the phenotype of TCPOBOP-treated Setd8-LKO cells, we do refer to the many other different features that have been proposed as consensus for senescent cells by the International Senescence Association (ICSA), reviewed in Ref. 3. These are:

1) Hallmark Feature 1: Cell cycle arrest. Setd8-LKO hepatocytes are clearly arrested in G2 phase, as evidenced by stopped growth and the activation of G2M-checkpoint genes (Supplementary Fig. 7d) and the induction of CDK2 inhibitor p21^{WAF1/Cip1} (CDKN1A) and CDK4/6 inhibitor p16^{INK4A} (CDKN2A) shown in Supplementary Fig 7e panel.

Although, we do detect Ki67 staining in about 19.4% of the large hepatocytes (Fig. 3c) and a number of EdU-positive cells (Supplementary Fig. 6a) after 24-hours of TCPOBOP treatment, this signal is highly reduced in the following days, indicating that DNA replication activity is restricted to the initial period leading to endoreduplication-mediated polyploidy, without cell division. Growth arrest is also supported by the absence of Ki67 signal in 80% of the cells, which already have reached the level of ploidy required for survival. Cell cycle arrest is also a general first effect of Setd8-deficiency in various established cell lines (Ref. 5-14).

Furthermore, at least for the period examined (up to 65 days), we did not observe normally dividing cells (by mitosis) even though the number of cells have increased. Our data demonstrate segregation of existing cells without further DNA replication and mitosis.

2) Hallmark Feature 2: Deregulated metabolism. The metabolic reprogramming in Setd8-LivKO hepatocytes has been described in our previous study (Ref. 16).

3) Hallmark Feature 3: Senescence Associated Secretory Phenotype (SASP). SASP is evident from the RNA data in Supplementary Fig. 7e.

4) Hallmark Feature 4: Macromolecular damage. This is demonstrated by the detection of γ H2AX (Fig. 4a) and also by the promiscuous binding of CTCF to several new regions (Supplementary Fig.10).

5) Other important features used to characterize senescence include:

a) SA- β -gal activity. In the revised version, we include close-up images of Setd8-deficient hepatocytes showing cytoplasmic SA- β -gal staining and also staining signal present in the intranuclear vesicles (Fig. 4b).

b) HMGB1 exit from the nucleus. In the revised version, we included an additional control panel from untreated Setd8-LKO livers (Fig. 4b). The concern that the observed export could be due to the chromatin changes mediated by Setd8-deficiency is excluded by the lack of HMGB1 export from the nucleus in untreated Setd8-LKO cells.

c) Senescence-associated heterochromatin foci (SAHF). In the revised Fig. 4c, we include macroH2A staining picture, which complements the evidence for the presence of such foci provided by the focal H3K9Me3-staining pattern in the original version.

d) Senescence Associated Stemness properties: This feature is demonstrated by the expression of oncofetal genes in TCBOBOP-treated Setd8-LKO livers (Supplementary Fig. 7c).

Reviewer 2. Minor

1. Diagram mentioned “micronuclei”. *Micronuclei specifically happens during mitosis, referring to the chromosome segregation error. Here the cells are not dividing and the structures are therefore not micronuclei.*

2. *MAP1LC3a mice are MAP1LC3B. LC3-GFP should be termed GFP-LC3.*

Response: We agree with the Reviewer for the erroneous use of the term “micronuclei”, which could be misleading. We have made the necessary text corrections in the text and the Figures describing the observed structures as “small nuclear detachment”.

We have also corrected the gene names as requested.

Reviewer 3. Point 1

The Atg5-dependency of the phenotype is interesting, providing some mechanistic insight, but the analysis is limited. E.g. in Sup. Fig. 1c, the increased nuclear area in setd8-LKO+TCPOBOP is partially reverted with Atg5 KO. Is this due to reduced intranuclear vesicle formation (i.e. hyperploidy still occurs), or is endoreduplication also inhibited with Atg5 deficiency? It would be important to clarify which aspects of the setd8-LKO+TCPOBOP phenotype are Atg5-dependent/Atg5-independent.

Response: To answer these questions, we have performed PI-staining assays and DNA-FISH experiments in TCPOBOP-treated Setd8/Atg5 double KO mice. As shown in the revised Supplementary Fig. 5c and d, polyploidization was significantly reduced. In the double KO mice, we could observe up to 16N hepatocytes as opposed to 64N and 128N chromosomal content in TCPOBOP-treated Setd8-LKO mice. This finding suggests that both hyperploidy and intranuclear vesicle formation depend on Atg5.

Taking into account that Atg5 is required for nuclear enlargement (as shown in Fig. 2f) and the findings that vesicle formation was completely eliminated, while polyploidization was still taking place albeit at a reduced scale in Setd8/Atg5 double KO mice, it is plausible to assume that hyperploidy is probably prevented indirectly due to the requirement of highly expanded nuclear space.

However, we cannot exclude a potential direct effect of Atg5 on nuclear DNA replication. In the revised manuscript, we refer to a study by (Maskey et al. Ref 29), which has shown that Atg5 can translocate to the nucleus in DNA damaged cells, where it binds to survivin complex and interferes with cytokinesis. However, it remains to be determined whether nuclear Atg5 may have other function, e.g. influencing DNA replication machinery. In this regard, we must note, that using a commercial antibody in preliminary experiments, we failed to detect nuclear Atg5 in the hepatocytes.

Reviewer 3. Point 2

Autophagy is typically cytoprotective but could also trigger cell death, particularly if its activity is exceptionally high. I wonder whether the massive cell death observed in the setd8-LKO+TCPOBOP liver is 'autophagic cell death'.

Response: These questions have been answered in our previous publications (Ref 15 and 16), where we showed that Setd8-LKO hepatocytes upon physiological or experimentally induced growth conditions, die via necrosis. We have also shown that unlike in cycling cell lines, where cell death occurs mainly

due to the requirement of Setd8 for mitotic DNA condensation and DNA repair, in normal G0 phase hepatocytes, Setd8-deficiency leads to gene expression alterations of key metabolic genes, which result in metabolic reprogramming. Among the metabolic changes, the altered ADP/ATP ratios activate AMPK, leading to increased fatty acid uptake, increased mitochondrial activity and parallel activation of the autophagy machinery, as evidenced by the detection of the lipidated form of LC3 and the detection of autophagosome structures by EM. As a result of the elevated mitochondrial activity, ROS production and accumulation is stimulated, which is further increased upon stress conditions. Therefore, in Setd8-deficient hepatocytes DNA-damage is initially induced by the elevated intracellular ROS levels and of course further aggravated by the defects in DNA repair process. The above order of events, suggest that the activation of the autophagy machinery in Setd8-LKO cells is initially cytoprotective (i.e. from excess lipid uptake). When ROS levels are elevated above a limit, AMPK-induced downstream pathways cannot compensate for the resulting DNA-damage effects and the cells are either eliminated by necrosis or enter a state displaying the senescence features described in this paper.

Reviewer 3. Point 3

"At 8 to 12 months of age, Setd8-deficient mice develop full-blown spontaneous liver tumors". What about setd8-LKO+TCPOBOP mice - are they more tumorigenic? Atg5 deficiency can be tumorigenic in the liver (e.g. PMID: 21498569). Do the TCPOBOP-treated Setd8/Atg5 double knockout mice show any acceleration of tumor development?

Response: The mechanism of the late-onset liver cancer phenotype in Setd8-LKO mice was beyond the scope of this paper.

We do have however, sufficient preliminary data showing that partial hepatectomy or mitogenic stimulus (TCPOBOP treatment) accelerated liver tumor formation, which can be detected as early as 5 months after birth (i.e. 3.5 months after treatment or operation).

In TCPOBOP-treated Setd8/Atg5 double KO mice, we observe tumor foci even earlier. The examples shown below are livers from 3.5 months old mice that were treated at day 45.

In our previous work (Ref 15), we have established, that the late-onset hepatocellular carcinoma cells are derived from transdifferentiating ductal progenitor cells, which express Setd8. In light of the current study, it will be interesting to see whether and how the SASP production or other signals from the surviving Setd8-LKO cells may contribute to this process.

WT

Setd8/Atg5-LKO + TCPOBOP

Reviewer 3. Point 4

I wonder if they can identify a similar phenotype in different contexts: how about e.g. setd8 ko fibroblasts with mitogenic stress in culture?

Response: We have not studied it and have no information about the effects of Setd8 deficiency in fibroblasts. With respect to the demonstration of endonucleosis in other conditions, our results are presented above in page 3-4 in response to Reviewer 2 Point 1.

REVIEWER COMMENTS

Reviewer #1 (Remarks to the Author):

The authors have well addressed my criticism on the frequency of the endonucleosis events.

However, the issue of the membrane composition of the endonucleosis vesicles has not been properly addressed and, in this Reviewer's opinion, is of very high importance. As the authors point out, Nup98 is a marker of both inner and outer nuclear membranes, so Nup98 positivity is not a proof of the presence of the outer nuclear membrane. Other antibodies against VAP-A or VAP-B, or ER proteins or immunoelectron microscopy should be able to clarify this issue

Reviewer #2 (Remarks to the Author):

The authors have satisfactorily addressed my previous critiques.

Reviewer #3 (Remarks to the Author):

I co-reviewed this manuscript with one of the reviewers who provided the listed reports. This is part of the Nature Communications initiative to facilitate training in peer review and to provide appropriate recognition for Early Career Researchers who co-review manuscripts

Reviewer #4 (Remarks to the Author):

While some questions are not addressed, the authors provide some nice preliminary data, and I agree that these are interesting for a future investigation. I have no further comments.

Response to Reviewer's Comment

Reviewer #1 has asked for additional experiments to clarify the involvement of the outer nuclear membrane in endonucleosis.

Reviewer #1 (Remarks to the Author):

The authors have well addressed my criticism on the frequency of the endonucleosis events. However, the issue of the membrane composition of the endonucleosis vesicles has not been properly addressed and, in this Reviewer's opinion, is of very high importance. As the authors point out, Nup98 is a marker of both inner and outer nuclear membranes, so Nup98 positivity is not a proof of the presence of the outer nuclear membrane. Other antibodies against VAP-A or VAP-B, or ER proteins or immunoelectron microscopy should be able to clarify this issue.

We followed the suggestion of the Reviewer and performed staining of liver tissues with different VAP-A antibodies using different fixation conditions and different secondary antibodies. As explained before, we expected staining the cytoplasmic membranes with some enrichment around the nuclear membrane. This was clearly the case in cultured cells (see Figure 1A below). In wild type liver tissue sections however, due to the thickness of the section, such differential cytoplasmic distribution of the signal is not evident (Figure 1B).

Importantly however, VAP-A signal was exclusively detected in the cytoplasm in wild-type cells and in the internal side of the endonucleotic vesicles in Setd8-LKO livers (Revised Supplementary Fig. 2e). Moreover, in focal panes we could also demonstrate denser signal near the inner vesicle membrane with gradual decrease towards more distant locations inside the vesicles.

To further strengthen this observation, we followed the Reviewer's suggestion and performed immunostainings with antibodies recognizing 3 different ER markers (PDI, Calnexin and ERp72). In all three cases, we detected immunostaining signals in the cytoplasm and the vesicle interior (Revised Supplementary Fig. 2f, 2g, 2h), where the distribution of the signals was similar to VAP-A.

These data, together with the Sun2 and Nup98 staining data, support the mechanism that nuclear envelope invaginations are of type II and that the process of endonucleosis involves both inner and outer nuclear membranes.

Figure 1

A (cultured cells)

VAP-A / DAPI

B (tissue section)

VAP-A / DAPI / Lamin AC

REVIEWERS' COMMENTS

Reviewer #1 (Remarks to the Author):

The Authors have satisfactorily addressed my criticism